# Mapping resilience: Development of the resilience process scales (RPS) and resilience profiles during adversity

**Joseph Anthony Pettit** *, **Stuart Beattie**, **Ross Roberts**, **Nichola Callow**

Institute for the Psychology of Elite Performance, Bangor University, Gwynedd, United Kingdom

☯ These authors contributed equally to this work.
* josephanthonypettit@gmail.com

## Abstract

The resilience literature is often criticised for lacking clarity in the conceptualisation and measurement of resilience, with the literature yet to consider within-person profiles of resilience and how such profiles might influence reactions to different adverse contexts. To significantly enhance the research area, the current set of studies propose and test a four-stage process model of resilience including proactive (anticipation & minimizing) and reactive (managing & mending) components. We suggest the four processes can function independently within five separate domains (general, physical, social, cognitive, and emotional). Specifically, in Studies 1 ($n = 181$) and 2 ($n = 284$) we develop a measure of resilience reflecting our four-stage process model and demonstrated validity of a 13-item measure for each of the five proposed domains via a Bayesian structural equation modelling approach. Focusing on the general domain and based on the four resilience processes (anticipate, minimize, manage, & mend), Study 3 ($n = 400$) explored resilience profiles in a pilot study, and then confirmed these profiles and their relationship with psychological and behavioral outcomes related to the COVID-19 pandemic in a main study. Using latent profile and latent transition analysis, results revealed four distinct profiles, predicting a range of psychological outcomes. For example, those with lower resilience (particularly profiles with high anticipation but low levels of the other processes), showed higher anxiety (especially with high anticipation), depression, impulsiveness, and lower coping effectiveness. Those with higher resilience (Profile 3 and 4) across the four processes exhibited lower depression, anxiety, and impulsiveness, as well as higher well-being, better perceived coping effectiveness, and preventative behaviors. Taken together the results from the studies presented, support the process model of resilience and underscore the benefits of considering resilience profiles in relation to understanding how people deal with adverse contexts.

**Data availability statement:** All relevant data are within the manuscript and its Supporting Information files. Raw data available at: https://osf.io/mtegk.

**Funding:** JP Funded by Knowledge Economy Skills Scholarships II, Outlook Expeditions Grant number: BUK2133 Websites. KESS II: https://kess2.ac.uk/ Outlook Expeditions: https://outlookexpeditions.com/. Outlook Expeditions supported in data collection for Study 1 and by extension, Study 3's pilot.

**Competing interests:** The authors have declared that no competing interests exist.

## Introduction

Successfully managing and adapting to life's challenges often requires resilience [1]. Resilience has been linked to coping better with mental ill-health, lower emotional distress following adversity, and faster recovery from such exposures [2,3]. Research in resilience tends to focus upon an individual's ability to manage with or recover from adversity. For example, the Connor-Davidson Resilience Scale (CD-RISC [4]) mainly focusses upon an individual's ability to manage adversity, whereas Smith et al.'s [5] Brief Resilience Scale focuses almost exclusively on recovery post-adversity. Some researchers propose that resilience is a stable trait (akin to an aspect of personality [6]) while others view resilience as a dynamic process via a combination of traits and states [7,8]. More recent research goes beyond the definition of resilience as just the ability to manage and recover from adversity, but rather as a process or series of processes.

### Resilience processes

Researchers such as Alliger et al. [9], Chen et al., [10], and Fletcher & Sarkar [11] have focused upon resilience as a process that can function within different specific domains. Alliger et al. [9] proposed a process model (albeit for team resilience), where processes of resilience contain the ability to proactively minimize (including both the ability to identify and appraise threats), manage (act and react, navigate, and then adapt as the issue occurs), and mend (react, learn, and recover) from the experience. Similarly, Chen et al. [10] proposed a more comprehensive three-stage process of resilience consisting of an individual's ability to anticipate (i.e., identify upcoming threat and being prepared), be flexible (adapt and manage), and bounce-back from adversity. Chen et al.'s model includes state and trait-like components of resilience, and the associated measure affords good psychometric properties. The processes of resilience also negatively relate to stress, anxiety, and depression.

While the work of Alliger et al. [9] and Chen et al. [10] represent advances in the study of resilience (i.e., they consider proactive behaviors), both approaches combine the processes of anticipating and minimizing into a single process (component). Under stress, most humans have an attentional bias towards threats, which usually has evolutionary benefits [12]. However, not every anticipated threat will require action and first requires an appraisal of one's potential and available coping resources [13]. For example, appraisal theory states that appraisals contain two main stages, primary appraisal, and secondary appraisals [14,15]. Primary appraisals require an individual to anticipate whether an upcoming situation is deemed as a threat, non-threat, or a challenge. If the individual perceives the situation as a threat, they will then engage with secondary appraisals (i.e., identifying resources available to minimize the threat). Here, individuals may employ early or additional coping strategies to minimize its potential impact (e.g., seek more information or seek help from a significant other). Or, as Farchi and Peled-Avram discuss, reframe these threats into challenges, and matching their available resources to the particular demands of the challenge to adapt and manage it [13]. However, if the individual anticipates the situation as a non-threat or a challenge, then they may make no behavioral adjustments.

In applied settings, Hardy et al. [16] found that athletes who demonstrated higher levels of performance when under pressure reported greater sensitivity to cues of potential punishment and threat in the environment. Hardy et al. also found further evidence to suggest that these athletes were predisposed to picking up threats early which, if equipped with appropriate coping strategies, allowed them more time to minimise and deal with such threats faster, leading to a better performance [16]. Therefore, given this differentiation between anticipate and minimize, theoretically it would seem prudent to consider resilience as having a proactive component containing two distinct processes (i.e., anticipate and minimize) as well as a reactive component also containing two distinct processes (i.e., manage and mend).

## Domains of resilience

Recent research further proposes that humans can be more resilient in certain domains of functioning than others. For example, Chen et al.'s [10] resilience model examines resilience within physical, emotional, and social domains of functioning (with examples such as maintaining better physical well-being and recovery, dealing with mental health problems such as depression, and maintaining better social support and interpersonal relationships). However, the authors do not clearly elaborate on how these domains were determined or how they were informed by previous research. Furthermore, there may be one more important domain of function missing from Chen et al.'s work that we term cognitive resilience. We discuss each of these domains in turn.

Regarding physical resilience, research has shown individuals with higher levels of resilience tend to cope better with physical illness and pain [17]. Other research has shown that athletes proactively modify their physical training loads to minimize the potential for physical stress and injury [18]. However, if resilience to physical pain/illness is low, to manage the negative emotions and emotional trauma that severe pain might bring, one would need to rely on a different type of resilience such as emotional resilience [19,20].

Experiencing adversity and stressful events generally causes negative emotional reactions, which can lead to negative outcomes on mental health [3]. Emotional resilience may offer a buffer against these negative responses, involving emotional regulation strategies to help appraise and reappraise before, during, and after an adverse event to deal with these emotions [21,22]. Schneider et al. [23] found that emotional regulating abilities and coping strategies facilitated resilience, leading to lower threat appraisals, and better management of positive and negative emotions when faced with the emotional stress that comes with adversity. Similarly, Stratta et al. [24] demonstrated that in the face of emotional trauma, emotional coping styles relate positively to an individual's resilience, which in turn acts as a protective factor from stress and can even guide toward a more positive outcome.

Social isolation, parental neglect, and interpersonal conflicts may also cause stress and lead to mental health problems if left unaddressed [25,26]. These problems can include aggression-eliciting effects from social exclusion [25], depression, anxiety, and posttraumatic stress disorder [26]. Wood and Bhatnagar [26] found that when facing social adversity, some individuals use active coping strategies (using one's resources to minimize impact) which are linked to higher resilience and better outcomes, whereas passive coping (e.g., avoiding or withdrawing) strategies are ineffective at dealing with social adversity [27]. Similarly, those with less resilience and who are more avoidant, engage in more problematic and addicted social media use [28]. Therefore, in today's first world society, given the perceived mental health risks associated with increased social media use in young adults [29], social resilience might be particularly salient.

Although Chen et al.'s [10] model acknowledges the above three domains of functioning, a final domain of importance is cognitive resilience. Cognitive resilience is demonstrated by an individual maintaining effective cognitive control under stress. Cognitive control can be compromised via reduced inhibitory control (e.g., inability to ignore distractions), reduced ability to shift attention between subtasks, and a reduction in the ability to update the working memory with new salient information [30]. Therefore, low cognitive resilience to stress and adversity can lead to poor performance, cognitive overload, burnout, and other mental health issues [31–33]. Research has demonstrated that stress from cognitive sources (such as cognitively demanding tasks) is dealt with differently to other sources of stress such as physical stress,

at a behavioral and physiological level [34,35]. Moreover, cognitive forms of stress can tax executive control systems in ways that can be different to more purely emotional forms of stress (which can often demand regulation of the feeling rather than information processing). For example, DeFraine [36] showed that cognitive load lowered emotional intensity only when participants passively experienced emotions, not when they deliberately maintained them – indicating that these two forms of emotional processing rely on distinct mechanisms. Indeed, there are a number of stressors, primarily cognitive in nature, that could impact an individual such as studying for an exam and completing assessments, which can cause cognitive stress resulting in lower performance and negatively affect mental health [37]. Second, one's job can provide a source of cognitive stress, for example air traffic controllers often experience high levels of cognitive pressure but maintain effective cognitive control [32]. Thus, resilience in the cognitive domain seems worthy of further investigation [38].

To summarise, alongside our proposed model of resilience as a four-stage process model with proactive and reactive components, we suggest resilience can be assessed at a general level and across more specific domains including physical, social, cognitive, and emotional. To the best of our knowledge, no measure currently exists that separates out the four processes of resilience across the five domains (including general resilience). Therefore, the purpose of Studies 1 and 2 was to develop and test an assessment tool that captures these processes and domains. We then used this measure to focus on the general domain of resilience in Study 3. The purpose of this final study was to provide an examination of resilience process profiles and how they related to behaviors and mental health outcomes during a unique and stressful experience (i.e., the COVID-19 pandemic).

## Study 1: Resilience process scales development and validation

### Method

**Resilience process item development.** Following a review of relevant literature and procedures on measurement development [39], the first part of the study involved creating an initial set of potential items assessing the resilience process across the four processes of anticipate, minimize, manage, and mend, alongside four domains. The initial research team included three academic psychologists with extensive expertise in resilience, sport and exercise psychology, and measurement development (the academic team) and a doctoral student. The academic team then assessed the content validity of these 65 items [39,40], where each item was examined as relating to its process (i.e., anticipate, minimize, manage, and mend), domain, and noted if items were deemed too similar in wording to another item or too complex on a 5-point Likert scale ranging from 1 (*not at all*) to 5 (*completely*) alongside verbal discussions in an iterative process [5,10,41,42]. We also presented these items to and utilized expert opinions from two outdoor expedition leaders and a workshop of other academic psychologists with over 100 years of combined experience. Such expert opinions can help ensure instruments are both meaningful and interpretable within applied settings [41].

This procedure led to a total of 20 items being retained with the conclusion that the items should be mapped only to each of the four resilience processes, with an instructional vignette to separate the domains of the same items [43]. Example items include "I can anticipate when a situation will stress me" (anticipate), "I tend to organise myself well to deal with challenges" (minimize), "When things get bad, I don't let them get to me" (manage), and "I bounce-back easily after a challenge" (mend). To assess resilience processes, we used a 7-point Likert-type response scale ranging from 1 (strongly disagree) to 7 (strongly agree). See S1 Appendix in supplementary materials for the initial 20 items.

Example items include "I can anticipate when a situation will stress me" (anticipate), "I tend to organise myself well to deal with challenges" (minimize), "When things get bad, I don't let them get to me" (manage), and "I bounce-back easily after a challenge" (mend). To assess resilience processes, we used a 7-point Likert-type response scale ranging from 1 (strongly disagree) to 7 (strongly agree).

**Domains of resilience.** Following a similar procedure to Cassidy [43], we developed instructional vignettes to separate domains of resilience into general, physical, social, cognitive, and emotional (see S2 Appendix in supporting

information for the final scales that include the vignettes). Using a general domain allows for an overall examination of the resilience process (i.e., anticipate, minimize, manage, and mend) free from context-specific domains. We used the same 20 items for each of the five domains, noting that the use of the vignette allows for them to be completed relative to each specific domain. We analysed the questionnaire using the Flesch-Kincaid algorithm [44] to ensure its potential utility amongst a broader population. We also ensured that individuals up to a reading level of grade 7.7 years ($M_{age}$ = 13) and above would be able to understand it.

The vignette presented for the general domain was as follows: "Please think of different tough situations, life events, challenges and obstacles that you may have experienced in the past or may experience in the future. This could be any event in which you have, or could experience stress, pressure, or hardship. Indicate below the extent to which you agree with each statement about yourself with regards to this/these experience(s), by circling the relevant number on the rating scale from "1" (Strongly Disagree) to "7" (Strongly Agree), with "4" being that you neither agree or disagree. Choose a number which best indicates your feelings about that statement. There are no right or wrong answers and all ratings will be kept confidential so please answer honestly. Everyone differs in how they deal with different situations. It is likely that there will be areas that you are better or worse in; this is totally normal and something we would expect." (see S2 Appendix in supporting information for the domain-specific vignette instructions).

**Participants.**  Following institutional ethical approval (Bangor University School of Psychology and Sport Science ethics review board approved each study from 2018), we recruited a convenience sample of 181 students ($M_{age}$ = 16.8 years, $SD$ = 0.74 years; $n$ = 95 Females, $n$ = 86 Male) from UK secondary schools and colleges who were about to depart on an overseas expedition (i.e., within 24 hrs of departure) and recruited with the support of the company Outlook Expeditions whom they were travelling with (see S3 Appendix in supplementary information for participant demographics). Participants provided written/digital consent between 13th June 2018–13th November 2018, with those under 16 also having parental/guardian consent granted prior to their participation. Overseas expeditions provide opportunities for individuals to experience physical, emotional, social, and cognitive adversity; thus we deemed this sample to have utility in relation to completing the measure.

**Measures and procedure.**  We used the 20-item Resilience Process Scales (RPS) reported above to assess resilience within each of the five domains of functioning (physical, emotional, social, cognitive, and general). We presented the vignettes in random order across participants to reduce the potential for order effects. Data in which participants were missing data, or put extreme scores across the domains [1 or 7 on every item] were removed.

Following the aforementioned institutional ethical approval, we contacted expedition leaders (including those used to help develop the items) and schoolteachers (to recruit their students) to inform them of this part of the study. To collect data from the students, the first author met them on their expedition's Final Preparation Day (FPD) at the expedition company's FPD centre. Students were already informed that a researcher was potentially conducting a study with them. The main researcher then explained the purpose of the study to each student and gained their informed consent. All students agreed to take part. Participants then completed a hard copy of the RPS which also included anti-social desirability instructions and assurance of anonymity.

**Analysis and model testing strategy.**  To test the factor structure of the RPS we used a Bayesian structural equation modelling (BSEM) approach [45] to factor analysis. The BSEM approach uses Bayesian inference to estimate relationships between observed and latent variables within a structural equation model. BSEM has several advantages over frequentist approaches such as Maximum Likelihood, and is increasingly advocated in the human sciences [46–48]. The BSEM approach views parameters as variables with a mean and distribution rather than constants, as in a maximum likelihood approach, allowing researchers to specify more realistic models and simultaneously allow small variances, cross-loadings, and correlated residuals within an identified model (leveraging past research or information) which results in more appropriate model fit statistics. In addition, BSEM is less influenced by smaller sample sizes and missing data (in contrast to other approaches) [45,48].

We analysed the factor structure for each domain of the RPS separately, thus repeated the following procedure for each of the five domains. Following contemporary methods [47,48], we first standardized the data at the item level and then estimated a series of three BSEM models. The first model incorporated noninformative priors for the major loadings, exact zero cross loadings and exact zero residual correlations. The second model incorporated the addition of informative approximate zero cross loadings. The final model incorporated the addition of both informative approximate zero cross-loadings and residual correlations. We specified the priors with a mean of 0 and a variance of.01. This size of prior corresponds to factor loadings and residuals with a 95% limit of ±.20, therefore representing small cross-loadings and correlated residuals [45,48].

We estimated all BSEM models with the Markov Chain Monte Carlo (MCMC) simulation procedure with a Gibbs sampler and a fixed number of 100,000 iterations for two MCMC chains to allow for the examination of model convergence. We assessed model convergence with the potential scale reduction factor (PSR). Model convergence is evident when the PSR value lies between 1.0 and 1.1 for all parameters [49]. In addition, we performed a visual inspection of trace plots for each parameter to check that the parameter values in each MCMC chain mixed well (i.e., converged to a similar target distribution; [50]). We assessed model fit using the posterior predictive p value (PPP value) and the symmetric 95% credibility interval for the difference between the observed and replicated χ2 values. A good-fitting model is indicated when PPP values are around.50 and credibility intervals centre on zero [45]. Once the final models were established, we performed a sensitivity analysis by rerunning the final models with variance priors specified at.005,.01, and.015 for the cross-loadings. We then examined parameter estimates to check for any important discrepancies, allowing us to understand the extent to which changing the priors influenced the stability of our estimates and model fit indices. We also examined interaction effects and differences between the resilience processes (see S3 Appendix in supporting information for these findings).

## Results

**Factorial validity.** All models converged adequately (except when adding a model using only cross-loadings for each domain without residual correlations; see Table 1). For all domains, the BSEM models with zero cross-loadings and zero residual correlations converged successfully, but the PPP for this model (with zero cross-loadings and residuals) indicated a poor fit to the data. The fit was also unacceptable for the models with informative small variance priors on the cross-loadings.

Across all domains however, models with informative small variance priors on the cross-loadings and residual correlations had a better fit to the data, with PPPs around.50 and symmetric 95% posterior predictive confidence intervals centred close to zero. While the initial 20-item models across each domain reported an acceptable model fit, there were items with relatively low standardized factor loadings, along with some of these items significantly loading above the tolerance set by the prior to one or more other items. These particularly problematic items were individually examined, assessed on the quality of the question in relation to the underlying construct, and removed where appropriate across each domain for consistency (see Table 2 for final list of items). This item removal process is common and accepted throughout measurement development, if removals are based on both theory as well as relevant data [46,51]. This process resulted in 13-item models across each domain (3 anticipate, 3 minimize, 4 manage, and 3 mend items), with good model fits evident, see S2 Appendix for the scales and scoring system.

Across each domain, all major loadings of the final model were significant. All cross-loadings were within the prior limits as were all but four correlated residuals. Upon further examination, there were no identifiable patterns between these four items, we therefore deemed it acceptable to keep these items as they were given their content validity, appropriate factor loadings and the otherwise acceptable model fit indices.

PSR values for the final models reached the 1.1 criterion at approximately 20,000 iterations for each domain. K-S tests for all parameters for both instruments were non-significant (p > .05). Visual inspection of the trace plots (162 parameters for each of the domains) all showed stability, with no upward or downward trends in the means and the two chains overlapping in variability.

**Table 1. Bayesian structural equation modelling (BSEM) fit statistics (N/A = non-convergence), including posterior predictive p values (PPP) and 95% credibility intervals (CI).**

| | BSEM Fit statistics | PPP | Difference between observed and replicated $\chi^2$ 95% CI | |
| --- | --- | --- | --- | --- |
| | | | Lower 2.5% | Upper 2.5% |
| General | 20-item Non-Informative | .000 | 209.92 | 312.41 |
| | 20-item Informative Priors (cross-loadings) | N/A | N/A | N/A |
| | 20-item Informative Priors (cross-loadings + residual correlations) | .59 | −68.95 | 53.90 |
| | 13-item Non-Informative | .000 | 39.17 | 110.77 |
| | 13-item Informative Priors (cross-loadings) | N/A | N/A | N/A |
| | 13-item Informative Priors (cross-loadings + residual correlations) | .54 | −43.78 | 38.87 |
| Physical | 20-item Non-Informative | .000 | 221.88 | 325.07 |
| | 1.20-item Informative Priors (cross-loadings) | N/A | N/A | N/A |
| | 2.20-item Informative Priors (cross-loadings + residual correlations) | .60 | −68.04 | 54.41 |
| | 13-item Non-Informative | .000 | 50.99 | 122.68 |
| | 3.13-item Informative Priors (cross-loadings) | N/A | N/A | N/A |
| | 4.13-item Informative Priors (cross-loadings + residual correlations) | .55 | −44.35 | 37.71 |
| Social | 5.20-item Non-Informative | .000 | 267.76 | 368.69 |
| | 6.20-item Informative Priors (cross-loadings) | N/A | N/A | N/A |
| | 7.20-item Informative Priors (cross-loadings + residual correlations) | .60 | −68.43 | 54.14 |
| | 13-item Non-Informative | .000 | 66.20 | 137.31 |
| | 8.13-item Informative Priors (cross-loadings) | N/A | N/A | N/A |
| | 9.13-item Informative Priors (cross-loadings + residual correlations) | .54 | −43.32 | 38.20 |
| Cognitive | 10.20-item Non-Informative | .000 | 227.00 | 330.89 |
| | 11.20-item Informative Priors (cross-loadings) | N/A | N/A | N/A |
| | 12.20-item Informative Priors (cross-loadings + residual correlations) | .60 | −68.56 | 54.59 |
| | 13-item Non-Informative | .02 | 4.11 | 75.56 |
| | 13.13-item Informative Priors (cross-loadings) | N/A | N/A | N/A |
| | 14.13-item Informative Priors (cross-loadings + residual correlations) | .55 | −43.86 | 39.12 |
| Emotional | 15.20-item Non-Informative | .000 | 272.00 | 373.19 |
| | 16.20-item Informative Priors (cross-loadings) | N/A | N/A | N/A |
| | 17.20-item Informative Priors (cross-loadings + residual correlations | .60 | −68.87 | 54.47 |
| | 18.13-item Non-Informative | .000 | 28.61 | 100.61 |
| | 19.13-item Informative Priors (cross-loadings) | .30 | −31.61 | 85.79 |
| | 20.13-item Informative Priors (cross-loadings + residual correlations) | .55 | −44.44 | 39.29 |

The estimated correlations between the four processes ranged between .32 to .69 in the general domain. For the physical domain, correlations between the four processes ranged between .51 to .69. For the cognitive domain, correlations ranged between .57 to .70. For the social domain, correlations ranged between .54 to .71. Finally, for the emotional domain, correlations ranged between .63 to .76. All of these relationships were significant and in the expected direction (see S1 Table in supporting information for these correlations). With regards to main differences, we also found significant effects between process, domain, and interaction (see S3 Appendix in supporting information for these findings).

Sensitivity analyses on the final models indicated that the factor loadings and cross-loadings were relatively stable when specifying prior variances for cross-loadings at smaller (.005) and greater (.015) values. However, the physical domain model would not converge on the smaller value prior, therefore it was run at 150,000 iterations in which no such

Table 2. BSEM standardized factor loadings of each item, including 95% CIs.

| | Standardized factor loadings for final items | Study 1 | | | | Study 2 | | | |
|---|---|---|---|---|---|---|---|---|---|
| | | Anticipate | Minimize | Manage | Mend | Anticipate | Minimize | Manage | Mend |
| General | I can anticipate when help is going to be needed. | **.69 [.25, 1.02]** | .02 [−.18,.21] | −.01 [−.20,.18] | −.01 [−.20,.19] | **.73 [.40, 1.00]** | −.01 [−.20,.17] | .02 [−17,.19] | −.01 [−.20,.19] |
| | I can anticipate when a situation will stress me. | **.65 [.13,.99]** | .02 [−.14,.18] | .04 [−.23,.15] | −.04 [−.23,.15] | **.82 [.50, 1.06]** | −.02 [−.20,.15] | −.05 [−.23,.13] | −.04 [−.23,.15] |
| | I notice possible difficult situations early. | **.70 [.33,.99]** | .00 [−.19,.18] | .05 [−.14,.24] | .06 [−.14,.25] | **.73 [.45,.99]** | .04 [−.14,.22] | .05 [−.14,.22] | .06 [−.14,.25] |
| | I make back-up plans for when things might go wrong. | −.02 [−.21,.17] | **.73 [.30, 1.02]** | .03 [−.22,.16] | −.04 [−.23,.15] | −.02 [−.20,.16] | **.82 [.51, 1.07]** | −.03 [−.21,.15] | −.04 [−.23,.15] |
| | I tend to organise myself well to deal with challenges. | −.02 [−.21,.16] | **.79 [.49, 1.04]** | .01 [−.17,.20] | .01 [−.18,.20] | −.02 [−.20,.14] | **.82 [.57, 1.04]** | .02 [−.16,.20] | .01 [−.18,.20] |
| | I prepare myself for upcoming challenges. | .05 [−.13,.23] | **.69 [.37,.97]** | .03 [−.16,.21] | .04 [−.15,.23] | .04 [−.14,.22] | **.70 [.40,.96]** | .02 [−.16,.20] | .04 [−.15,.23] |
| | I remain positive, even when things seem hopeless. | .01 [−.17,.18] | .01 [−.17,.19] | **.77 [.44, 1.17]** | −.01 [−.20,.18] | −.01 [−.18,.16] | −.00 [−.18,.17] | **.78 [.49, 1.05]** | −.01 [−.20,.18] |
| | When things get bad, I don't let them get to me. | −.04 [−.22,.14] | −.04 [−.22,.15] | **.78 [.41, 1.17]** | .02 [−.18,.22] | .02 [−.15,.20] | .02 [−.16,.19] | **.75 [.40, 1.02]** | − .02 [−.18,.22] |
| | I keep a clear head under pressure. | .04 [−.15,.23] | .02 [−.17,.21] | **.52 [.03,.88]** | .02 [−.18,.22] | −.01 [−.19,.17] | −.02 [−.20,.26] | **.56 [.42, 1.04]** | .02 [−.18,.22] |
| | I give my best effort no matter the obstacle. | .01 [−.18,.20] | .05 [−.14,.25] | **.55 [.08,.93]** | −.01 [−.21,.19] | .03 [−.17,.21] | .04 [−.15,.24] | **.50 [.08,.90]** | −.01 [−.21,.19] |
| | I bounce-back easily after a challenge. | −.00 [−.18,.18] | .01 [−.18,.18] | .03 [−.18,.22] | **.81 [.49, 1.07]** | −.01 [−.18,.17] | .02 [−.17,.20] | .02 [−.19,.22] | **.78 [.46, 1.07]** |
| | I quickly get over set-backs. | −.01 [−.18,.19] | −.03 [−.21,.15] | .04 [−.17,.23] | **.77 [.40, 1.06]** | −.01 [−.19,.17] | −.04 [−.22,.13] | .03 [−.18,.23] | **.82 [.48, 1.09]** |
| | I know how to stop the same things getting to me in the future. | −.01 [−.17,.20] | −.04 [−.15,.23] | −.04 [−.24,.16] | **.71 [.33, 1.03]** | .05 [−.15,.24] | .04 [−.15,.22] | −.02 [−.22,.18] | **.73 [.40, 1.04]** |
| Physical | I can anticipate when help is going to be needed. | **.74 [.38, 1.08]** | .00 [−.19,.19] | .03 [−.17,.23] | .00 [−.19,.19] | **.75 [.42, 1.06]** | .02 [−.18,.22] | .02 [−.18,.20] | .02 [−.19,.21] |
| | I can anticipate when a situation will stress me. | **.82 [.49, 1.09]** | −.01 [−.19,.17] | −.02 [−.21,.17] | −.01 [−.20,.18] | **.77 [.39, 1.07]** | −.04 [−.23,.14] | −.05 [−.24,.13] | −.07 [−.25,.12] |
| | I notice possible difficult situations early. | **.74 [.39, 1.05]** | .02 [−.17,.20] | .01 [−.19,.20] | .02 [−.17,.21] | **.74 [.46, 1.02]** | .03 [−.16,.22] | .014 [−.15,.22] | .06 [−.14,.25] |
| | I make back-up plans for when things might go wrong. | −.03 [−.16,.21] | **.79 [.49, 1.07]** | .02 [−.17,.20] | .01 [−.18,.20] | −.01 [−.17,.19] | **.78 [.50, 1.06]** | .02 [−.17,.20] | .02 [−.17,.21] |
| | I tend to organise myself well to deal with challenges. | .03 [−.15,.22] | **.79 [.49, 1.06]** | −.01 [−.19,.17] | .01 [−.18,.20] | −.01 [−.20,.16] | **.89 [.62, 1.15]** | −.01 [−.20,.17] | −.01 [−.19,.18] |
| | I prepare myself for upcoming challenges. | **−.05 [−.23,.13]** | **.82 [.48, 1.10]** | .01 [−.18,.19] | −.00 [−.20,.23] | −01 [−.18,.20] | **.79 [.48, 1.09]** | .01 [−.19,.19] | −.01 [−.19,.18] |
| | I remain positive, even when things seem hopeless. | .05 [−.16,.25] | .03 [−.17,.19] | **.53 [.12,.91]** | .04 [−.16,.23] | **−.02 [−.20,.16]** | .02 [−.17,.21] | **.79 [.47, 1.08]** | .01 [−.19,.20] |
| | When things get bad, I don't let them get to me. | .01 [−.19,.21] | .01 [−.22,.15] | **.74 [.40, 1.10]** | .01 [−.18,.20] | −.01 [−.20,.17] | −.01 [−.20,.18] | **.72 [.39, 1.04]** | .00 [−.20,.21] |
| | I keep a clear head under pressure. | .01 [−.18,.20] | −.01 [−.19,.17] | **.82 [.51, 1.09]** | −.00 [−.19,.18] | .01 [−.16,.20] | −.01 [−.20,.18] | **.76 [.46, 1.05]** | .01 [−.19,.21] |
| | I give my best effort no matter the obstacle. | −.03 [−.22,.17] | −.01 [−.20,.18] | **.70 [.31, 1.03]** | −.01 [−.21,.17] | .02 [−.17,.20] | .00 [−.19,.19] | **.74 [.41, 1.05]** | .00 [−.20,.20] |
| | I bounce-back easily after a challenge. | .00 [−.19,.19] | .04 [−.17,.24] | −.01 [−.21,.18] | **.71 [.28, 1.05]** | −.01 [−.19,.16] | −03 [−.22,.16] | −.00 [−.20,.19] | **.87 [.57, 1.16]** |
| | I quickly get over set-backs. | −.01 [−.20,.18] | −.03 [−.23,.16] | .02 [−.18,.21] | **.79 [.44, 1.08]** | −.02 [−.20,.16] | −.01 [−.20,.17] | .03 [−.18,.22] | **.82 [.51, 1.11]** |

*(Continued)*

| | Standardized factor loadings for final items | Study 1 | | | | Study 2 | | | |
|---|---|---|---|---|---|---|---|---|---|
| | | Anticipate | Minimize | Manage | Mend | Anticipate | Minimize | Manage | Mend |
| | I know how to stop the same things getting to me in the future. | −.01 [−.18,.20] | .02 [−.18,.21] | .01 [−.18,.20] | **.79 [.47, 1.08]** | .05 [−.14,.23] | .05 [−.16,.24] | .01 [−.20,.21] | **.75 [.42, 1.08]** |
| Social | I can anticipate when help is going to be needed. | **.77 [.43, 1.08]** | .02 [−.18,.20] | .02 [−.18,.22] | .01 [−.18,.20] | **.77 [.46, 1.05]** | .02 [−.18,.21] | .02 [−.17,.20] | .02 [−.17,.21] |
| | I can anticipate when a situation will stress me. | **.80 [.43, 1.11]** | −.03 [−.21,.16] | .00 [−.20,.20] | −.02 [−.21,.17] | **.77 [.44, 1.04]** | −.01 [−.20,.17] | −.03 [−.21,.15] | −.03 [−.22,.16] |
| | I notice possible difficult situations early. | **.82 [.53, 1.09]** | .02 [−.16,.20] | −.00 [−.20,.19] | .02 [−.16,.19] | **.78 [.47, 1.06]** | .02 [−.17,.20] | .02 [−.17,.21] | .03 [−.17,.22] |
| | I make back-up plans for when things might go wrong. | .06 [−.21,.17] | **.76 [.49, 1.05]** | .04 [−.15,.22] | .01 [−.18,.19] | −.01 [−.19,.18] | **.85 [.58, 1.09]** | .03 [−.16,.21] | −.02 [−.20,.16] |
| | I tend to organise myself well to deal with challenges. | −.03 [−.21,.16] | **.88 [.59, 1.15]** | −.08 [−.26,.11] | .01 [−.18,.19] | .01 [−.18,.19] | **.78 [.47, 1.05]** | .01 [−.18,.21] | .02 [−.18,.20] |
| | I prepare myself for upcoming challenges. | **−**.02 [−.13,.23] | **.82 [.52, 1.09]** | −.06 [−.14,.25] | −.00 [−.19,.18] | .03 [−.17,.21] | **.78 [.47, 1.05]** | −.01 [−.20,.20] | .03 [−.16,.21] |
| | I remain positive, even when things seem hopeless. | **−**.02 [−.22,.18] | −.03 [−.22,.16] | **.77 [.43,1.01]** | −.00 [−.19,.18] | **−**.01 [−.19,.18] | .01 [−.17,.20] | **.81 [.50, 1.08]** | .03 [−.18,.22] |
| | When things get bad, I don't let them get to me. | .02 [−.19,.22] | .03 [−.16,.22] | **.74 [.40, 1.05]** | .02 [−.17,.21] | −.01 [−.18,.18] | .01 [−.18,.19] | **.72 [.39, 1.01]** | .04 [−.16,.24] |
| | I keep a clear head under pressure. | .04 [−.17,.23] | .01 [−.18,.20] | **.62 [.23,.97]** | .03 [−.17,.22] | .00 [−.18,.18] | −.00 [−.19,.18] | **.78 [.46, 1.06]** | −.01 [−.20,.18] |
| | I give my best effort no matter the obstacle. | .00 [−.20,.20] | .01 [−.18,.20] | **.67 [.27, 1.01]** | −.01 [−.20,.18] | .02 [−.17,.21] | .01 [−.19,.21] | **.61 [.23,.96]** | −.02 [−.22,.18] |
| | I bounce-back easily after a challenge. | −.02 [−.21,.18] | .02 [−.18,.21] | −.01 [−.21,.19] | **.81 [.41, 1.09]** | .00 [−.17,.18] | .01 [−.18,.18] | .01 [−.19,.19] | **.85 [.59, 1.12]** |
| | I quickly get over set-backs. | −.01 [−.21,.18] | −.03 [−.22,.17] | .03 [−.17,.22] | **.77 [.19,.98]** | −.00 [−.18,.17] | −.04 [−.22,.13] | .05 [−.16,.24] | **.80 [.50, 1.08]** |
| | I know how to stop the same things getting to me in the future. | .04 [−.17,.23] | .02 [−.18,.22] | .00 [−.19,.20] | **.71 [.36, 1.11]** | .02 [−.17,.21] | .05 [−.15,.24] | −.03 [−.22,.17] | **.77 [.45, 1.07]** |
| Cognitive | I can anticipate when help is going to be needed. | **.73 [.32, 1.07]** | .02 [−.18,.21] | .03 [−.17,.23] | .02 [−.18,.21] | **.73 [.42, 1.02]** | .00 [−.19,.19] | .04 [−.15,.22] | .01 [−.17,.20] |
| | I can anticipate when a situation will stress me. | **.79 [.40, 1.10]** | −.02 [−.21,.17] | −.03 [−.22,.17] | −.01 [−.20,.19] | **.82 [.52, 1.06]** | −.01 [−.20,.18] | −.04 [−.22,.15] | −.02 [−.21,.16] |
| | I notice possible difficult situations early. | **.77 [.42, 1.09]** | .02 [−.17,.21] | .02 [−.19,.22] | .01 [−.19,.20] | **.81 [.56, 1.05]** | .01 [−.17,.20] | .01 [−.18,.19] | .01 [−.17,.20] |
| | I make back-up plans for when things might go wrong. | .06 [−.13,.24] | **.82 [.57, 1.08]** | .06 [−.13,.23] | −.00 [−.19,.18] | −.00 [−.19,.17] | **.82 [.52, 1.11]** | .00 [−.19,.19] | .01 [−.19,.20] |
| | I tend to organise myself well to deal with challenges. | −.03 [−.22,.15] | **.87 [.60, 1.12]** | −.05 [−.22,.13] | .03 [−.16,.21] | −.00 [−.18,.17] | **.77 [.46, 1.05]** | .01 [−.18,.20] | .01 [−.18,.19] |
| | I prepare myself for upcoming challenges. | **−**.02 [−.21,.16] | **.89 [.61, 1.15]** | −.01 [−.19,.17] | −.01 [−.20,.17] | .00 [−.18,.18] | **.79 [.47, 1.07]** | .00 [−.19,.19] | .01 [−.18,.19] |
| | I remain positive, even when things seem hopeless. | .03 [−.18,.23] | .01 [−.18,.20] | **.66 [.29,.99]** | .02 [−.18,.22] | **−**.00 [−.18,.17] | .01 [−.18,.20] | **.81 [.52, 1.09]** | −.01 [−.21,.18] |
| | When things get bad, I don't let them get to me. | .02 [−.18,.22] | .05 [−.15,.24] | **.71 [.35, 1.04]** | −.00 [−.20,.19] | .01 [−.17,.19] | −.01 [−.19,.18] | **.70 [.33, 1.00]** | .02 [−.18,.23] |
| | I keep a clear head under pressure. | −.00 [−.20,.19] | −.02 [−.21,.16] | **.75 [.03, 1.07]** | .02 [−.18,.22] | −.04 [−.22,.14] | −.02 [−.21,.16] | **.81 [.51, 1.10]** | .03 [−.18,.22] |
| | I give my best effort no matter the obstacle. | −.02 [−.22,.18] | −.02 [−.22,.18] | **.73 [.32, 1.08]** | −.01 [−.21,.19] | .06 [−.13,.26] | .07 [−.15,.27] | **.55 [.16,.92]** | −.00 [−.20,.20] |
| | I bounce-back easily after a challenge. | −.03 [−.22,.17] | .01 [−.19,.21] | −.03 [−.22,.17] | **.80 [.39, 1.13]** | −.01 [−.19,.16] | .02 [−.17,.21] | .00 [−.20,.20] | **.81 [.50, 1.09]** |

*(Continued)*

**Table 2.** (Continued)

| | Standardized factor loadings for final items | Study 1 | | | | Study 2 | | | |
|---|---|---|---|---|---|---|---|---|---|
| | | Anticipate | Minimize | Manage | Mend | Anticipate | Minimize | Manage | Mend |
| | I quickly get over set-backs. | .05 [−.15,.25] | .03 [−.17,.22] | .02 [−.18,.22] | **.64 [.25,.99]** | −.02 [−.19,.17] | −.04 [−.23,.15] | .03 [−.18,.23] | **.80 [.47, 1.09]** |
| | I know how to stop the same things getting to me in the future. | .00 [−.19,.19] | −.02 [−.21,.17] | .03 [−.17,.23] | **.86 [.54, 1.16]** | .04 [−.14,.21] | .03 [−.17,.22] | .01 [−.21,.19] | **.80 [.50, 1.08]** |
| Emotional | I can anticipate when help is going to be needed. | **.72 [.36, 1.02]** | .03 [−.17,.22] | .04 [−.16,.24] | .02 [−.18,.22] | **.81 [.51, 1.07]** | .00 [−.19,.18] | .04 [−.16,.24] | .01 [−.18,.19] |
| | I can anticipate when a situation will stress me. | **.65 [.54, 1.17]** | −.04 [−.23,.14] | −.01 [−.20,.18] | −.05 [−.24,.14] | **.85 [.58, 1.08]** | −.03 [−.21,.14] | .01 [−.18,.19] | −.03 [−.21,.14] |
| | I notice possible difficult situations early. | **.78 [.48, 1.08]** | .03 [−.16,.22] | −.01 [−.20,.17] | .04 [−.16,.22] | **.67 [.37,.95]** | .05 [−.14,.23] | −.04 [−.22,.14] | .05 [−.14,.23] |
| | I make back-up plans for when things might go wrong. | .03 [−.15,.21] | **.79 [.50, 1.10]** | .02 [−.17,.20] | .03 [.17,.23] | −.01 [−.19,.17] | **.82 [.56, 1.07]** | .05 [−.13,.23] | .00 [−.19,.19] |
| | I tend to organise myself well to deal with challenges. | −.01 [−.19,.17] | **.88 [.56, 1.16]** | −.02 [−.21,.16] | −.02 [−.22,.17] | −.02 [−.20,.16] | **.86 [.61, 1.09]** | .01 [−.18,.19] | .00 [−.19,.18] |
| | I prepare myself for upcoming challenges. | **−.03 [−.21,.16]** | **.83 [.51, 1.12]** | .01 [−.18,.19] | .00 [−.20,.19] | .04 [−.15,.22] | **.76 [.46, 1.04]** | .00 [−.19,.18] | .01 [−.19,.19] |
| | I remain positive, even when things seem hopeless. | .01 [−.19,.21] | .01 [−.19,.21] | **.66 [.25, 1.02]** | .01 [−.19,.20] | **−.02 [−.19,.16]** | .01 [−.18,.18] | **.82 [.53, 1.10]** | .01 [−.19,.21] |
| | When things get bad, I don't let them get to me. | .04 [−.17,.24] | .07 [−.14,.26] | **.64 [.23, 1.01]** | .05 [−.15,.25] | .03 [−.20,.15] | .01 [−.17,.18] | **.77 [.44, 1.07]** | .02 [−.18,.22] |
| | I keep a clear head under pressure. | −.00 [−.19,.18] | −.03 [−.22,.15] | **.75 [.40, 1.05]** | −.01 [−.20,.18] | .05 [−.16,.21] | −.03 [−.22,.15] | **.78 [.45, 1.07]** | .00 [−.20,.20] |
| | I give my best effort no matter the obstacle. | −.03 [−.22,.17] | −.02 [−.12,.18] | **.76 [.36, 1.09]** | −.03 [−.22,.17] | −.03 [−.16,.24] | .04 [−.17,.23] | **.53 [.14,.93]** | −.01 [−.21,.19] |
| | I bounce-back easily after a challenge. | .01 [−.19,.20] | .04 [−.18,.24] | −.01 [−.21,.18] | **.76 [.39, 1.11]** | −.03 [−.21,.15] | .04 [−.15,.20] | .02 [−.19,.22] | **.84 [.56, 1.11]** |
| | I quickly get over set-backs. | .02 [−.19,.22] | −.02 [−.22,.18] | .04 [−.17,.24] | **.72 [.33, 1.08]** | .01 [−.17,.18] | −.03 [−.21,.14] | .02 [−.18,.22] | **.84 [.55, 1.12]** |
| | I know how to stop the same things getting to me in the future. | −.01 [−.20,.19] | −.04 [−.20,.20] | .00 [−.20,.20] | **.82 [.47, 1.16]** | .04 [−.15,.23] | .02 [−.17,.20] | −.02 [−.22,.18] | **.72 [.37, 1.04]** |

PPP = posterior predictive *p* value; BSEM = Bayesian Structural Equation Modelling. Factor loadings and 95% credibility intervals in bold correspond to the items in each row.

issues occurred [50]. Across each domain, more than 99% of discrepancies were within ±.05, with the maximum discrepancy across all sensitivity analyses being.06. Taken together, the minimal change in parameter estimates across the different analyses provides support for the stability of the scales.

**Internal consistency.** Composite reliability coefficients of the anticipation, minimize, manage, and mend subscales demonstrated acceptable reliabilities between.76 to.90 (see S3 Appendix for the full list of composite reliability scores).

## Discussion

The results of Study 1 provide initial support for the RPS, with the modified 13-item scales showing acceptable factorial validity from the BSEM analyses. It is noteworthy that the sample in Study 1 contained a group of young adults preparing for an expedition. Therefore, we set out to confirm the factor structure of the RPS in a more heterogeneous sample in Study 2, allowing us to generalise our findings beyond this initial sample.

## Study 2: Scale confirmation

### Method

**Participants.** Following institutional ethical approval (Bangor University School of Psychology and Sport Science ethics review board approved each study from 2018), we recruited a convenience sample of 284 participants from community and University based groups ($M_{age}$ = 26.4 years, $SD$ = 10.5 years; $n$ = 139 Female, $n$ = 144 Male) via social media, posters, and face-to-face recruitment (see S3 Appendix in supporting information for further participant demographics). Participants provided written/digital consent prior to their participation.

**Measures and procedure.** We used the final stage BSEM model from Study 1. Data in which participants had missing data were removed, and those that put extreme scores across the domains [1 or 7 on every item] were removed from final analyses.

### Results

When allowing priors on cross-loadings and correlated residuals all models converged (reaching the 1.1 PSR criteria at approximately 20,000 iterations) and showed good fit to the data. K-S tests for all parameters for each instrument were non-significant, and visual inspection of trace plots showed stability. Item factor loadings were all significant and above accepted limits (see Table 2). Correlations ranged between .40 and .83, with similar significances and direction to Study 1 (see S2 Table in supporting information for these correlations). Sensitivity analysis performed on the final models showed that factor loadings and cross-loadings were relatively stable when specifying prior variances for cross-loadings at smaller (.005) and greater (.015) values. Similar to Study 1, across each domain, 97% of discrepancies were within ±.05, with the maximum discrepancy across all sensitivity analyses being .07. Regarding mean differences, again similar to Study 1 we also found significant effects between processes and domains and also obtained a process × domain interaction (see S4 Appendix in supporting information for these findings).

### Discussion

The results of Study 2 provide further confirmatory support for the proposed resilience scales in a more heterogeneous population. The analysis demonstrated an acceptable model fit with good factor loadings of the final 13-item scale within each domain. These factor loadings were consistent with Study 1 in terms of magnitude.

With the initial factorial validity of the RPS established, there is a need for the process model of resilience to be tested. Indeed, the current model should allow for more insightful examination of an individual's resilience. That is, by examining profiles of the four processes, we would be able to consider *where* an individual is relatively 'high' or 'low' on a particular resilience process, and what the impacts of particular resilience profiles are on a host of dependent variables. In Study 3 we conducted a pilot study to explore and gain a theoretical foundation of what common resilience profiles might emerge, and what they might look like. Then in the main part of Study 3, we examined resilience profiles and the relationships that the profiles had on various outcomes in relation to a substantive and pervasive stressor, the COVID-19 pandemic, in addition to the stability and influence of these variables over time.

## Study 3: Latent resilience profiles, psychological and behavioral effects, and the COVID-19 pandemic

The COVID-19 pandemic caused significant psychological and physical adversity, characterised by social isolation, negative impacts on physical health, education, and finances, and led to increasing levels of stress, depression, and anxiety, as well as reduced levels of well-being and resilience in the population [52–54]. The emotional distress attached to the isolation may have been worsened by the limited availability of usual social support and routines that acted as coping strategies [55]. Furthermore, these multiple sources of adversity relating to the pandemic may have had a compounding effect with more typical daily stressors, compromising resilience throughout daily life [56], and as such, the pandemic provides a unique context of adversity in which to investigate resilience.

## Pilot study

In the pilot study, we tested what preliminary profiles emerge from the proposed resilience processes. As this part of the multi-study paper represents the application of a novel approach to resilience, and, for the purposes of brevity, we decided to take an overarching view by focusing on the general domain. However, we did initial profile testing on the other four domains, revealing similar findings and profile patterns, these results can be received on request to the first author.

Given the novelty of the process model, it is difficult to offer strong hypotheses as to what profiles of resilience might be expected. However, based on the research literature, we did make initial theoretical predictions resulting in the expectation that the following four general profiles would emerge: a profile where both proactive and reactive components of resilience were relatively low, a profile where the proactive components were higher than the reactive, a profile where reactive components were higher than proactive, and then a profile with all processes relatively high (for detail on our hypotheses, see S5 Appendix in supporting information). We pilot tested the appropriateness of our initial theorising using the samples from Study 1 and 2, and an additional convenience sample of 90 students (providing written/digital prior to their participation, with a recruitment period from 18th April 2019–18th January 2020, those under 16 also had parents or guardians indicate written/digital consent prior to their participation; Bangor University School of Psychology and Sport Science ethics review board approved each study from 2018) from secondary schools and sixth forms across the UK (we had a combined sample of 555; see S5 Appendix, S3 Table, & S1 Fig in supporting information for further detail on this study; see Analysis section in the Main Study below for the profile analysis approach taken; data in which participants were missing data, or put extreme scores across the domains [1 or 7 on every item] were removed). Results revealed four distinct profiles that were somewhat consistent with our predictions. We describe these profiles below.

Based on the literature and the pilot study findings (see S5 Appendix, S3 Table, & S1 Fig in supporting information), we demonstrated the following profiles and hypothesise the following outcomes from them in relation to the COVID-19 pandemic. Note that these profiles are titled in relation to their relative mean scores on the RPS (i.e., Profile 1 has anticipate as a higher scored process relative to minimize, manage, and mend).

Profile 1 *Low Resilience – High Anticipate*: Profile 1 (7.4% of our sample in the pilot study) consisted of a lower level of resilience, but with anticipate higher than the other processes. Given the correlations between severity of COVID-19 experience, well-being, and resilience, with resilience having a protective effect on well-being [52,57], it seems likely that individuals in this profile may experience increased stress, anxiety, and depression during the pandemic. A high level of anticipation with lower levels in the other processes may be associated with over-active threat sensitivity coupled with low coping responses, leading to high levels of anxiety [58–60]. Further, low levels of managing and mending may also lead anxiety due to effects such as over-rumination [61]. Negative behavioral outcomes such as impulsive behavior [62–64] may also manifest as a way of attempting to cope.

Profile 2 *Lower Resilience – High Proactive*: This profile (32.8% of our sample in the pilot) consisted also of lower resilience, but with high anticipate and minimize strategies. We expected that these individuals would likely present similar outcomes to Profile 1, particularly regarding depression and well-being. Behaviorally, they would likely often prepare for things well in advance, taking minimizing actions such as precautionary or preventative measures (e.g., mask-wearing) that would reduce negative affect and physical risk [65,66]. However, as individuals with lower ability to manage and mend, may present as taking fewer risks and have less effective coping methods [67,68], struggling to deal well with or recover from unexpected events they do not perceive themselves as able to control.

Profile 3 *Moderate Resilience – High Reactive*: This profile (45.6% of our sample in the pilot) has similar levels of each process with moderate resilience, but with slightly elevated levels of manage and mend. Individuals in this profile would be expected to report lower levels of depression and anxiety than Profile 1. A higher manage and mend in comparison to the other processes may suggest more effective coping methods and behaviors [69]. The relatively lower levels of anticipate

and minimize compared to manage may be associated with more frequent risk-taking behaviors. High risk-taking may be due to a high perceived ability to cope and handle various threats [67] such as those associated with the pandemic, and reflect less perceived need to be aware of and plan for these threats.

Profile 4 *High Resilience – High Reactive*: This profile (14.2% of our sample in the pilot) had high resilience across all processes, but with a relatively higher amount of manage and mend. Having a high level of resilience, these individuals would be expected to be far less negatively affected by the pandemic, with a greater sense of well-being and more effective coping reported [70], lower levels of depression and anxiety, and present fewer undesirable behaviors around COVID-19 [52,57,71]. With a greater anticipate, these individuals may be more inclined to perceive and consider threats to a greater degree [72] during the pandemic, but may also take more calculated risks with the perception of being able to manage and mend effectively from the potential threat [61,64,67].

### Main study

In the main study, and employing a new sample, we examined the extent to which the profiles revealed in the pilot did indeed predict psychological-related outcomes relating to the COVID-19 pandemic, namely anxiety, depression, well-being, and behavioral-related outcomes (including risk-taking, impulsiveness, undesirable coping behaviors, and preventative behaviors). We also examined these same outcomes four months later from a pool of this new sample. Further, we explored the stability and reliability of the emerging resilience profiles, in which these profiles are expected to be relatively stable [7], with participant's membership to their respective profile to mostly stay the same over time. Gaining insight into these outcomes could help inform on the psychological impact of the pandemic and the role of resilience in these outcomes. It may also develop our understanding of resilience having state or trait-like characteristics, as well as guide future interventions in improving psychological and behavioral outcomes.

### Method

**Participants.** Following institutional ethical approval (Bangor University School of Psychology and Sport Science ethics review board approved each study from 2018), we recruited a new, convenience sample of 400 participants ($M_{age}$ = 32.1, $SD$ = 8.9; $n$ = 184 Male, $n$ = 183 Female, $n$ = 33 preferred not to say) via social media (see S6 Appendix in supplementary information for participant demographics). Data collection was conducted during the UK pandemic lockdown (beginning February 2021). Participants provided digital consent prior to their participation, with a recruitment period from 14th January 2021–13th July 2021.

To examine the stability of resilience profiles across the pandemic, approximately four months after the original data collection we invited participants to complete the measures for a second time. At this second data collection point, 175 of the original 400 participants responded ($M_{age}$ = 31.3, $SD$ = 7.6; $n$ = 84 Male, $n$ = 77 Female; $n$ = 14 preferred not to say).

### Measures

**Resilience.** We used the General domain of the RPS (see Study 1 & 2). Data in which participants were missing data, or put extreme scores across the domains [1 or 7 on every item] were removed.

**Depression and anxiety.** We measured depression and anxiety using the 4-item Patient Health Questionnaire (PHQ-4 [73]). Items are anchored on a 3-point Likert-type scale following the statement "Over the last 2 weeks, how often have you been bothered by the following problems?" (1 = *not at all* to 3 = *nearly every day*). Factor analysis by Kroenke et al. [73] showed good validity along with Cronbach's alpha scores all over .80. Although the four items are usually measured together, given the item's origins in the PHQ-2 and Generalized Anxiety Disorder (GAD-2), we examined them separately [74]. Both the PHQ-2 and GAD-2 have shown good criterion and convergent validity, sensitivity to change, and good internal consistency (*a* = .83 & .86, respectively [75,76]. We therefore chose to score depression and anxiety in this PHQ-2 and GAD-2 format, respectively.

**Well-being.**  We used the World Health Organisation's (WHO) recommended COVID-19 survey tool [77] to examine the psychological outcomes of well-being. The WHO-5 is amongst the most widely used questionnaires assessing subjective psychological well-being and has been found to have adequate validity in screening for depression and in measuring outcomes in clinical trials [78]. Five items are anchored on a 5-point Likert-type scale (5 = *all of the time* to 1 = *at no time*) with the statement "Over the past 2 weeks…" followed by items such as "I have felt cheerful and in good spirits". Previous studies report Cronbach's alpha scores from.83 to.92 [79].

**Coping effectiveness.**  We assessed perceptions of coping effectiveness using the 7-item Coping Effectiveness Scale [80,81]. The items are anchored on a 4-point Likert type scale (1 = *strongly disagree* to 4 = *strongly agree*). The instruction given to participants was reworded to ask how they felt they were coping with the pandemic, with example items such as "I'm dealing with this problem better now than I used to" (referring to the stresses related to the pandemic) followed by the Likert-type scale. Convergent validity of the scale has been demonstrated in positive relationships with positive framing and affect, with Cronbach's alphas ranging from.66 to.74 [80,81].

**Impulsiveness.**  We measured impulsiveness using the Barratt Impulsiveness Scale – Brief [82]. This scale comprises eight items anchored on a 4-point Likert-type scale (1 = *rarely* to 4 = *almost always/always*) with items such as "I do things without thinking". Cronbach's alpha scores from previous studies ranged between.83 and.86 and provided evidence of reliability [82].

**Risk-taking.**  We measured risk-taking using the General Risk Propensity Scale [83], an 8-item measure anchored on a 5-point Likert-type scale (1 = *strongly disagree* to 5 = *strongly agree*) that examines risk-taking as a general personality disposition with items such as "I am attracted, rather than scared, by risk". The measure shows acceptable discriminant and convergent validity and internal consistency [83].

**Preventative behaviors.**  The WHO recommended COVID-19 survey tool [77] also has measures we used to examine the outcome of preventative behaviors – covering a range of actions one might take to protect from and prevent infection (such as wearing a mask in public). Preventative behaviors measured on are a 9-item inventory anchored on a 7-point Likert type scale (1 = *not at all* to 7 = *very much so*; followed by *not applicable [which were discounted from any mean score]*), with items adapted from Steel-Fisher et al. [84] and including "Wore a mask in public" and "I frequently washed my hands with soap and water for at least 20 seconds". Following factor analysis to validate these measures (see S6 Appendix in supporting information), a good model fit was found (PPP of .50; CI of −20.50 and 21.21), with significant factor loadings of .41, .55, .60, .52, .57, and .55 for the final six items used.

## Procedure

To examine the stability of resilience profiles across the pandemic, participants were sent the aforementioned measures to complete in their own time online (being presented in random order to mitigate for order effects). These data were then used to examine what types of profiles exist and how many, which were then used to examine different outcomes across our dependent variables. This was then followed by gathering this data again at the second timepoint. These data points are hereafter referred to as Timepoint 1 and Timepoint 2.

## Analysis

**Latent profile analysis.**  We first conducted latent profile analysis (LPA) to identify subgroups of individuals based on their responses to the Resilience Process Scale's general domain. LPA is a latent variable approach to identifying subgroups within a population based on a set of variables (in our case, the four resilience processes) and predicted number of profiles. Compared to similar methods such as cluster analysis, LPA allows more flexibility in model specification and provides users with several fit indices to assess the quality of model fit, making it a superior approach to other person-centred clustering methods [85]. While there is no "gold standard" for determining the optimum number of profiles, it is generally worthwhile exploring a range of predicted profile solutions, starting with a lower number (1 profile) and building up from here when running the analysis [85]. We examined solutions from one to six latent

profiles in which the means and variances of the resilience processes were freely estimated in all profiles. Then we tested beyond our predicted four profiles to examine if a more complex model offered a better fit to our data than a more parsimonious one.

To determine the ideal number of profiles in the data, multiple factors should be considered, including the substantive meaningfulness (including a meaningful group membership size within a profile), theoretical conformity, sample size, and statistical adequacy [86,87]. To support decision making, LPA offers several statistical indices including the Akaike's Information Criterion (AIC), Bayesian Information Criterion (BIC), and Adjusted BIC (aBIC) values. Comparing each proposed model (number of profiles), these statistics measure the trade-off between fit and complexity (i.e., accuracy and over-fitting – with over-fitted models being less generalizable and less likely to replicate), with a lower relative score reflecting a better model fit. An entropy score is also given, where a higher relative score represents a clearer delineation of profiles. Further, the analysis also provides $p$ values of the Lo-Mendell-Rubin likelihood ratio test (LMR LRT) and Bootstrap LRT (BLRT). These $p$ values compare the currently examined model to the previous (the model with one fewer profiles), with a significant value indicating the current model is a better fit.

**Exploring effects of profiles on outcomes.** To examine the impact of profiles on the psychological and behavioral outcomes, we used an extension of the LPA method in which auxiliary variables can be added using the DU3STEP command in Mplus [88], which allows for the examination of the influence of resilience profiles on the psychological and behavioral outcomes. We initially ran these analyses on the data from Timepoint 1. Following this first set of analyses, we then explored the influence of resilience profiles at Timepoint 1 on outcomes at Timepoint 2 using the data from those who had completed measures at both timepoints.

**Latent transition analysis.** Finally, to examine the stability of resilience profiles across time, we conducted two separate LPAs, followed by latent transition analysis (LTA; [86]). LTA is an extension of LPA in which one can estimate the probabilities of transitions among profiles over time based on the likelihood-ratio $G^2$ statistic, in addition to AIC, BIC, and entropy scores. The two LPAs help establish that profiles remain the same at each timepoint. If this is found, LTA (assuming the profile structures themselves are identical or very similar at each timepoint) is used to examine if members of a resilience profile tend to stay in their profile over time, or if their profile membership changes, and if so, to which profile they transition to. This prediction is made via maximum likelihood estimation, producing a posterior probability (how strongly individuals are associated with the estimated profiles) of participant's patterns of responses at each time point,

The sample sizes for LPA and LTA are similar to other research that use LPA, latent class analysis (LCA) and LTA [86,89], and appropriate for a sample size of at least 100 participants [90]. In addition, we also screened for whether negative affect [77] and past COVID infection influenced resilience profiles (see S6 Appendix in supporting information).

## Results

**Resilience profiles.** The process of LPA resulted in a four-profile solution that was relatively consistent with our pilot findings and hypotheses, with some small differences. Table 3 shows model fit indices with the 4-class solution demonstrating lower AIC, BIC, and BIC, a good entropy value (.78), significant LMR LRT and BLRT scores ($p < .05$), and meaningful group sizes. The fit indices demonstrated a more optimal fit with four profiles in comparison to other solutions. The entropy score also showed a clearer delineation of profiles, along with an acceptably high group membership within each profile (e.g., each profile had a minimum of 10 participants). A 5-class solution also indicated a good model fit. However, three of the profiles in the 5-class solution had a relatively low group membership (11, 19, and 24, from a sample of 400) making this solution impractical to use going forward, alongside a LMR LRT score that was not a significantly better fit than 4-class. For these reasons, in addition to the practical limitations of our sample size, and consistency with the pilot study, we used a four-profile model. In other populations and sample sizes, other numbers of profiles may produce better fits.

Fig 1 contains the standardized pattern of anticipate, minimize, manage, and mend for each of the four profiles, along with the number of participants contained in each respective profile. We named Profile 1 (n = 49; 12.3% of the sample)

**Table 3. Fit indices comparing different profile solutions.**

| LPA Outputs | AIC | BIC | Adj BIC | Entropy | LMR LRT | BLRT |
|---|---|---|---|---|---|---|
| 1 Class | 4552.60 | 4584.53 | 4559.15 | N/A | N/A | N/A |
| 2 Class | 4275.79 | 4327.68 | 4286.43 | 0.750 | <.001 | <.001 |
| 3 Class | 4232.51 | 4304.36 | 4247.24 | 0.762 | <.001 | <.001 |
| 4 Class | 4191.89 | 4283.70 | 4210.71 | 0.778 | 0.003 | <.001 |
| 5 Class | 4175.78 | 4287.54 | 4198.70 | 0.825 | 0.213 | <.001 |

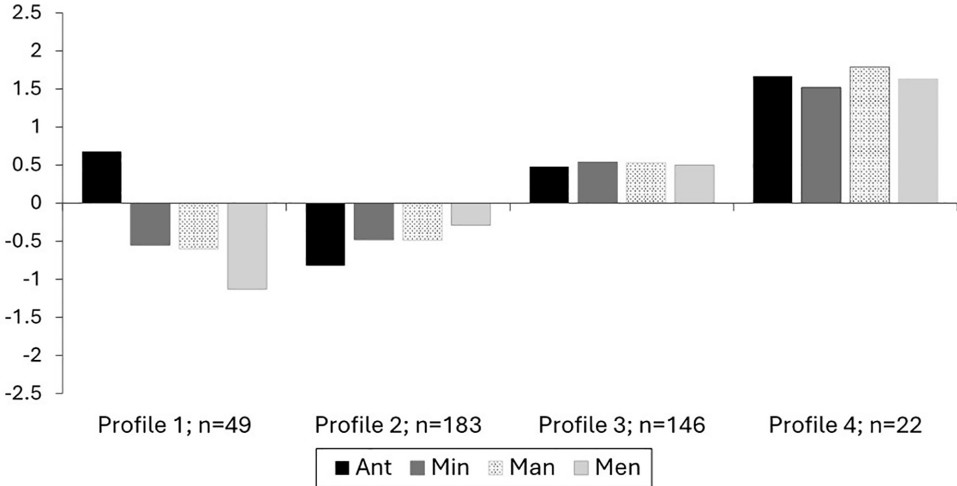

**Fig 1. Study 3 standardised profiles (n = 400) by Profile 1 (Low Resilience – High Anticipate), Profile 2 (Low Resilience – Low Anticipate), Profile 3 (Moderate Resilience), and Profile 4 (High Resilience).**

*Low Resilience – High Anticipate.* Individuals in this profile reported a low level of resilience, with anticipate being higher and mend being relatively lower as hypothesised.

We named Profile 2 (n = 183; 45.8% of the sample) *Low Resilience – Low Anticipate.* Individuals in this profile reported a low level of resilience, with anticipate being particularly low. Though not directly predicted, this profile is similar to our hypotheses, but with the anticipate process being low rather than high.

We named Profile 3 (n = 146; 36.5% of the sample) *Moderate Resilience.* Individuals in this profile reported a moderate level of a resilience, with similar levels of each process. This profile is also similar to our hypothesised Profile 3.

Lastly, we named Profile 4 (n = 22; 5.5% of the sample) *High Resilience*, which a small amount of the sample belonged to. Individuals in this profile reported a high level of resilience across the board, similar to the hypothesised Profile 4. A degree of caution should be used when interpreting outcomes from this profile due to the smaller sample size limited. This profile also emerged in the pilot study, but with a larger group size but still a relatively small percentage of the total sample (likely due to the larger total sample size). Below are the reported outcomes variables from our measures for each of these four resilience profiles (see Table 4 for all outcome means across profiles).

## Impact of resilience profiles

**Anxiety.** The profiles demonstrated significant differences in anxiety ($\chi2 = 13.51$, $p = .004$). Specifically, Profile 1 ($M = 4.47$) displayed the highest score, and Profile 4 the lowest ($M = 2.73$). Members of Profile 1 (*Low Resilience –*

**Table 4. Mean (M) outcome scores across each profile.**

| | Profile 1 Low Res – High Ant | | Profile 2 Low Res – Low Ant | | Profile 3 Mod Resilience | | Profile 4 High Resilience | |
|---|---|---|---|---|---|---|---|---|
| | *M* | *SE* | *M* | *SE* | *M* | *SE* | *M* | *SE* |
| Anxiety | 4.47 | 0.23 | 3.69 | 0.10 | 4.30 | 0.16 | 2.73 | 1.03 |
| Depression | 4.50 | 0.25 | 3.74 | 0.08 | 3.98 | 0.11 | 2.70 | 0.25 |
| Well-Being | 45.06 | 2.61 | 47.12 | 0.83 | 45.58 | 1.80 | 68.76 | 6.54 |
| Risk-Taking | 2.79 | 0.14 | 3.19 | 0.03 | 3.59 | 0.07 | 3.42 | 0.30 |
| Impulsiveness | 17.92 | 0.76 | 19.60 | 0.26 | 17.18 | 0.35 | 12.46 | 1.05 |
| Coping Effectiveness | 2.46 | 0.09 | 2.49 | 0.03 | 2.46 | 0.05 | 3.04 | 0.11 |
| Preventative Behavior | 4.94 | 0.20 | 4.85 | 0.07 | 5.65 | 0.09 | 6.14 | 0.23 |

*High Anticipate*) and 3 (*Moderate Resilience*) were significantly higher in anxiety than Profile 2 (*Low Resilience – Low Anticipate*; $M_{diff}=0.77$, $p=.005$; $M_{diff}=0.61$, $p=.002$ respectively). However, although members of Profile 4 (*High Resilience*) had the lowest mean score for anxiety, this profile was not significantly different to any other profile. This was most likely due to a relatively very high standard error ($SE=1.03$) in anxiety scores for Profile 4, alongside its limited sample size.

**Depression.** The profiles differed in depression ($\chi2=29.75$, $p<.001$), with Profile 1 ($M=4.50$) displaying the highest score. Depression scores were greater for those in Profile 1 than Profile 3 ($M=3.98$, $M_{diff}=0.53$, $p=.063$), and significantly higher than Profile 2 ($M=3.74$, $M_{diff}=0.76$, $p=.006$), and Profile 4 ($M=2.70$, $M_{diff}=1.80$, $p<.001$). Depression scores were also significantly greater for those in Profile 2 and Profile 3 compared to Profile 4 ($M_{diff}=1.03$, $p<.001$; $M_{diff}=1.27$, $p<.001$ respectively).

**Well-being.** There was a significant difference across profiles for well-being ($\chi2=11.85$, $p=.008$), with Profile 4 ($M=68.76$) displaying the highest score, and Profile 1 ($M=45.06$) the lowest. Profile 4 was associated with significantly higher well-being than each other profile (with a minimum $M_{diff}=21.64$, and a maximum $p=.001$). No other differences were significant.

**Risk-taking.** We observed differences in risk-taking across the profiles ($\chi2=36.66$, $p<.001$). Specifically, Profile 3 ($M=3.59$) had the highest levels of risk-taking and Profile 1 ($M=2.79$) the lowest. Risk-taking scores for those in Profile 3 were significantly higher than Profile 1 ($M_{diff}=0.80$, $p<.001$), and Profile 2 ($M=3.19$, $M_{diff}=0.41$, $p<.001$). Those in Profile 1 were also significantly lower in risk-taking than Profile 2 ($M_{diff}=0.40$, $p=.006$), and lower in risk-taking (with the difference approaching significance) than Profile 4 ($M=3.42$, $M_{diff}=0.63$, $p=.067$).

**Impulsiveness.** The profiles showed significant differences in impulsiveness ($\chi2=71.45$, $p<.001$), with Profile 2 ($M=19.60$) displaying the highest score and Profile 4 ($M=12.46$) the lowest. Profile 2 had significantly higher levels of impulsiveness than Profile 1 ($M_{diff}=1.68$, $p=.048$), Profile 3 ($M_{diff}=2.42$, $p<.001$), and Profile 4 ($M_{diff}=7.14$, $p<.001$). Those in Profile 4 were also significantly lower than Profile 1 ($M_{diff}=5.47$, $p<.001$), and Profile 3 ($M_{diff}=4.72$, $p<.001$).

**Coping effectiveness.** The profiles showed significant differences in coping effectiveness ($\chi2=23.69$, $p<.001$), with those in Profile 4 ($M=3.04$) having the highest score and Profile 1 ($M=2.46$) the lowest. Members of Profile 4 were significantly higher than those in Profile 1 ($M_{diff}=0.59$, $p<.001$), Profile 3 ($M_{diff}=0.59$, $p<.001$), and Profile 2 ($M_{diff}=0.55$, $p<.001$). No other profiles significantly differed.

**Preventative behavior.** The profiles showed significant differences in preventative behavior ($\chi2=70.12$, $p<.001$), with Profile 4 ($M=6.14$) displaying the highest score, and Profile 2 ($M=4.85$) the lowest. Those in Profile 1 engaged in significantly fewer preventative behaviors than Profile 3 ($M_{diff}=0.71$, $p=.001$) and Profile 4 ($M_{diff}=1.20$, $p<.001$). Those in Profile 2 engaged significantly less than Profile 3 ($M_{diff}=0.71$, $p<.001$) and Profile 4 ($M_{diff}=1.29$, $p<.001$). The difference in engaging in more preventative behaviors between those in Profile 4 and Profile 3 approached significance ($M_{diff}=0.49$, $p=.051$).

**Causal outcomes of resilience profiles.** As a supplement to the cross-sectional analysis above, and to investigate the potential longitudinal effects of resilience using the data from participants who had completed all measures at both time points (*n* = 175), we examined the influence of resilience profiles at Timepoint 1 on outcome variables at Timepoint 2 (we also examined resilience profiles at Timepoint 2 with Timepoint 2 outcomes, producing similar findings to those presented here with Timepoint 1 profiles). In these analyses there were no differences across the profiles for anxiety, coping effectiveness, and preventative behaviors. However, we found differences between profiles for the other outcome variables and in most cases, the nature of the differences between profiles were similar to the cross-sectional analyses (see S7-S8 Appendix in supporting information for these findings).

**Profile stability and transitions.** In addition to the potential psychological and behavioral outcomes of the four resilience profiles, collecting data at multiple timepoints also allowed us to examine the stability of the profiles in understanding to what extent members of one profile might transition into another across this time frame. LPAs at each timepoint using a four-profile solution (see Figs 2, 3; Table 5 for fit indices), showed a good model fit in both instances to use this same 4-class solution for LTA. Profiles remained relatively similar in pattern and magnitude. However, profile membership did change somewhat that highlighted the need for transition analysis to examine this further. Transition analysis demonstrated similar findings of good model fit and demonstrated some stable profiles and some relative instabilities ([86]; see Table 6 & Fig 4).

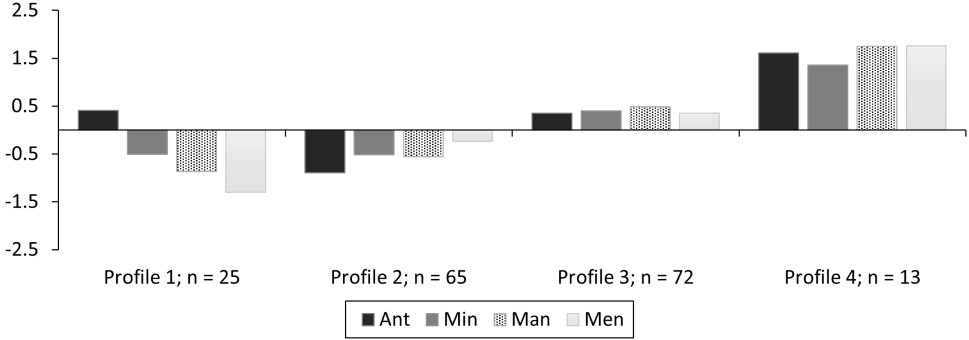

**Fig 2. Study 3 standardized profiles (n = 400) for Timepoint 1 by Profile 1 (Low Resilience – High Anticipate), Profile 2 (Low Resilience – Low Anticipate), Profile 3 (Moderate Resilience), and Profile 4 (High Resilience).**

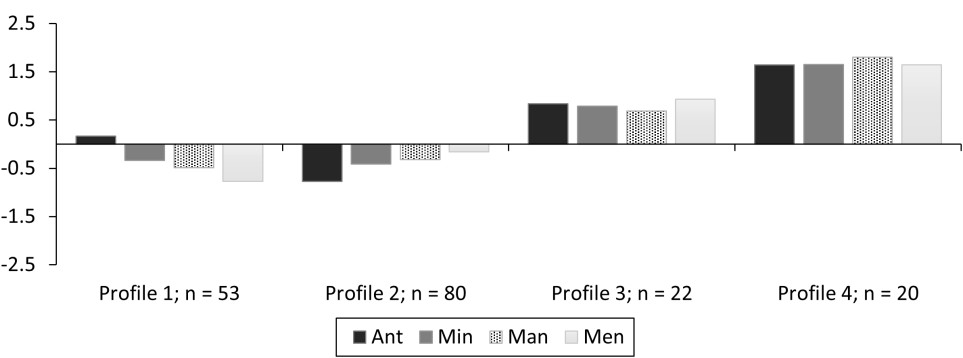

**Fig 3. Study 3 standardized profiles (n = 175) for Timepoint 2 by Profile 1 (Low Resilience – High Anticipate), Profile 2 (Low Resilience – Low Anticipate), Profile 3 (Moderate Resilience), and Profile 4 (High Resilience).**

**Table 5. Fit indices comparing LPA model fit at each timepoint.**

| LPA Outputs | AIC | BIC | Adj BIC | Entropy | LMR LRT | BLRT |
|---|---|---|---|---|---|---|
| Time 1: 4 Class | 1817.94 | 1890.72 | 1817.89 | 0.775 | 0.223 | <.000 |
| Time 2: 4 Class | 1708.26 | 1781.05 | 1708.22 | 0.727 | 0.726 | 0.03 |

**Table 6. Fit indices for LTA model fit, followed by profile transitions by each column (in bold represents their stability).**

| LTA Outputs | AIC | BIC | Adj BIC | Entropy |
|---|---|---|---|---|
| 4 Class | 1817.94 | 1890.72 | 1817.89 | 0.804 |
| | **Profile 1** | **Profile 2** | **Profile 3** | **Profile 4** |
| Profile 1 – Low Resilience – High Anticipate | **0.803** | 0.132 | 0.020 | 0.115 |
| Profile 2 = Low Resilience – Low Anticipate | 0.001 | **0.744** | 0.497 | 0.312 |
| Profile 3 – Moderate Resilience | 0.090 | 0.058 | **0.351** | 0.001 |
| Profile 4 – High Resilience | 0.106 | 0.066 | 0.132 | **0.573** |

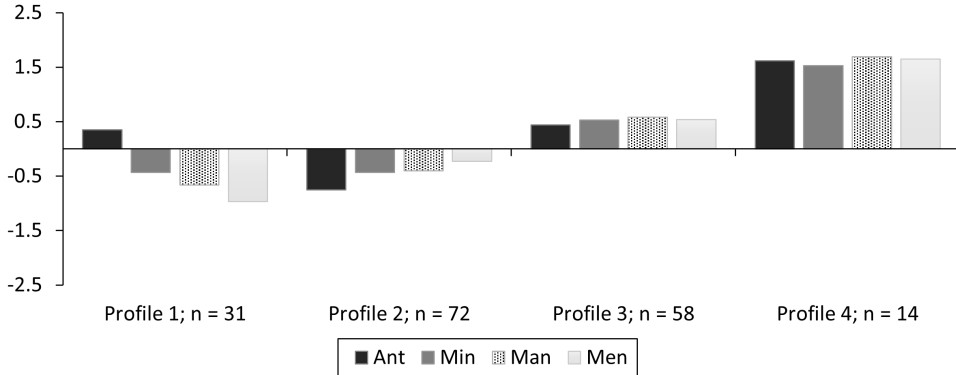

**Fig 4. Standardized profiles (n = 175) of LTA over the two timepoints by Profile 1 (Low Res – High Ant), Profile 2 (Low Res – Low Ant), Profile 3 (Mod Resilience), and Profile 4 (High Resilience). Group membership is at Timepoint 1.**

Profile 1 (*Low Resilience – High Anticipate*; n = 31) was stable (80.3% of participants remained in this profile across time) with small transitions into Profile 3 (*Moderate Resilience*; n = 58; 9%) and Profile 4 (*High Resilience*; n = 14; 10.6%), with transition probabilities (see Table 6) demonstrating a shift from lower resilience to higher resilience over the timepoints. Profile 2 (*Low Resilience – Low Anticipate*; n = 72) was also stable (74.4%) with small transitions into Profile 3 (5.8%), Profile 4 (6.6%), and Profile 1 (13.2%), showing some participants shifting from lower resilience to higher, and some moving to a similar profile with higher anticipation but lower mending. Profile 3 had relative instability (35.1%) with transition probabilities demonstrating a shift into Profile 2 (49.7%) and Profile 4 (13.2%), showing those with moderate resilience tended to shift to lower resilience or higher resilience over the timepoints. Profile 4 was somewhat stable (57.3%) with transitions into Profile 1 (11.5%) and Profile 2 (31.2%), showing those with high resilience shifting to lower resilience profiles. To summarise, LTA showed that depending on the profile, resilience is not necessarily stable and can change for some people across time.

## General discussion

The purpose of the current set of studies was to provide and test a new four-stage process model for resilience including proactive (anticipation & minimizing) and reactive (managing & mending) components within five domains of functioning.

After developing the resilience process scales, we explored profiles of resilience based on the four processes. We then examined the influence of the profiles on different behavioral and psychological outcomes during an adverse context (COVID-19 pandemic), as well as their stability over a four-month period.

## Resilience as a process of processes

**Proactive resilience.** The studies support previous research which puts forward proactive components to resilience [9–11]. Moreover, the present studies extend the previous research by providing evidence for the importance of separating the anticipation and minimizing processes (via their distinctions in the BSEM and resilience profiles). Anticipate refers to awareness of upcoming threats and adversity, and minimizing refers to actions taken to prepare for and reduce the impact of adversity. In Studies 1 and 2, anticipation was consistently rated higher than minimizing, managing, and mending (see S3 and S4 Appendix in supporting information). The results suggest that some individuals are very aware of upcoming potential stressors and threats, and that they may or may not be required to act upon them [91]. However, those with high proactive strategies but low reactive strategies may have an attentional bias towards threats which they may or may not be very good at dealing with should they fail to minimize them [92].

**Reactive resilience.** The studies also support (also via the components' distinctions in the BSEM and resilience profiles) the idea that managing and bouncing back is an essential *part* of what encapsulates resilience. Reactive resilience becomes more salient when individuals fail to anticipate threats or lack the resources to deal with them adequately. This definition fits with the more traditional approaches to resilience [5,8] before its conceptualisation evolved and expanded [9–11]. Current results would support the importance of both proactive *and* reactive strategies. However, future research should continue to validate the existence of these four processes of resilience, alongside their potential antecedents and outcomes. In addition, further research could examine the timing sequence of proactive and reactive resilience. Generally speaking, we present resilience as having proactive aspects that precede reactive. However, some situations such as prolonged adversity and coping, or contexts in which events could only be reacted to (such as an emergency) and not anticipated or planned for could trigger these processes in different orders (e.g., starting with manage, and then onto mend, and even then looping back to manage). Researchers could investigate these types of contexts and adversities, and how they may influence the timing and order of the resilience processes, profiles, and outcomes.

## Domains of functioning

Another purpose of the current studies was to examine resilience across five different domains of functioning (e.g., general, physical, social, cognitive, and emotional). In Studies 1 and 2, emotional was the lowest rated domain of resilience, and physical as the highest. This difference may reflect how general populations from these groups navigate and experience these types of adversity. For example, physical forms of stress such as illness, fatigue, pain etc. can be more tangible and recognisable issues (for example, physical pain is one of the most reported problems in medical care [92]) so may be more regularly dealt with. But more importantly, physical stress tends to have more obvious solutions over psychological issues (e.g., rest, medication, or see a doctor), whereas psychological issues may be more cognizant upon emotional awareness and regulation [93,94] which future research could aim to investigate.

In the introduction we made the case for cognitive resilience as an important yet untapped domain for resilience, with support found for this domain via the BSEM of Study 1 and 2. Recognising and understanding how one can be resilient to varying cognitive stressors could be vital in situations and jobs where performance relies upon complex cognitive task performances (e.g., air traffic control; [32]). Neglecting to deal with cognitive stressors such as time pressures or financial issues may have a detrimental effect upon attentional regulation and working memory [30]. This in turn may lead to mental health issues [37,38] and poor task performance that resilience could protect against [95]. However we recognise our investigations and this theorising reflect initial considerations related to cognitive resilience, that require validation. Therefore,

future research could use our measurement tool to examine cognitive resilience across different aspects of life in order to both further validate the domain, as well as use it as a potential predictor of performance under cognitive forms of stress.

## Psychometric contributions

The psychometric methods, properties, and strengths have been briefly mentioned in Study 1, given some of their limitations it is worth discussing some implications to both general measurement development and to resilience measures. First, all studies proposing and developing measures should state clearly and concisely the conceptualisation of the construct they are measuring. Historically, this has been lacking in the resilience literature [8] as well as providing good psychometric evidence for measures, as we attempt to do in the current studies. Furthermore, our proposed measure examines resilience as a dynamic process that includes consideration for the context and kinds of adversity faced (e.g., domains), as well as the proactive and reactive nature of resilience as discussed above, which few resilience measures have previously set out to do. Most current measures of resilience instead have generally focused on outcomes related to resilience, or its protective qualities (particularly traits [8,96]. For example, the Resilience Scale [97] focuses on personal factors of resilience (Personal Competence & Acceptance of Self and Life), whereas our measure includes consideration to behaviors ("I make back-up plans for when things might go wrong") and social resources ("I can anticipate when help is going to be needed"), that are also generally considered a part of resilience [7,98]. Though it should be noted that the presented studies similarly rely on self-reports of resilience, limiting validity when not combined or triangulated with observed or peer-reported ratings.

## Resilience profiles

Generally, the profiles scoring on extreme ends of the continuum such as very high and very low resilience had fewer members, perhaps supporting a normal distribution. However, a degree of caution should be exercised interpreting comparisons with the high resilience profile due to its limited sample size (though it should be noted that a small proportion of the sample in the pilot study also belonged to a similar profile but with higher numbers – likely in part due to a larger total sample size). Most of our participants fit into a profile that was somewhat low to moderate in resilience, with differences across the proactive and reactive components of the model being evident. The findings of Study 3 also extend resilience research by examining within-person profiles whereby the different components of resilience are examined concurrently across time. This within-person approach allows for a more nuanced way of measuring trait and state aspects of resilience whereby some stability would support a trait argument.

However, some participants were subject to change. For example, where those with high resilience transitioned to a lower resilience profile (and vice versa). Fluctuations in resilience may have been due to some of the participants catching coronavirus for the first or indeed a second time, or an effect of the prolonged stress over this period [99]. Depending on how well they coped with their symptoms, may have had an indirect influence in their perceptions of resilience (and/or physical resilience). This finding supports previous research that resilience is a combination of both state and traits factors (particularly in being state-like) [7,8]. Furthermore, it may also reflect how these perceptions and self-reports may also be dependent on their potential or actual coping resources available to them (and if they are recognised or used) [13]. Future research may want to more closely monitor potential adversities across time and investigate how antecedents such as resources, negative affect, or illness may relate to the transitions across resilience profiles themselves rather than just profile memberships at a singular moment.

In addition, we did find in the main study of Study 3 a Low Resilience – Low Anticipate profile that did not emerge in the pilot. It may have been that during the main study (during lockdown), participants had fewer threats they would readily recognise when (for example) unable to leave their home. Furthermore, a significant portion of the pilot sample were students of a younger age and perhaps feel more awareness anticipation of threats and adversity that they face (e.g., an upcoming expedition or university studies). Future research could examine which profiles emerge in different populations and contexts.

 

## Profiles and psychological outcomes

Generally, a higher level of proactive and reactive resilience was associated with better psychological outcomes such as lower depression, anxiety, and more appropriate behaviors such as taking more preventative measures during the pandemic [65,66]. In addition, high levels of anticipation along with the ability to minimize, manage, and mend effectively was associated with more risk-taking, low impulsiveness, and more positive outcomes. The ability to anticipate threats may be both a benefit and a hindrance, where it is advantageous only if there is capacity to actually deal with perceived adversities. This may indeed lead to more calculated but low impulsive risk-taking, which in adverse environments could lead to greater reward and experience – nurturing further resilience [11,69,100].

Research also highlights that coping strategies tend be seen as more effective if an individual perceives they have a high level of resilience. For example, Beattie et al. [101] found that in a sample of ultra-marathon runners, the relationship between coping effectiveness and performance satisfaction was moderated by reactive resilience. Specifically, they found that if the athletes were equipped with strong reactive processes (i.e., manage and mend resilience strategies), then coping effectiveness had a significant positive relationship with performance satisfaction. However, under conditions of low reactive resilience, coping effectiveness had a non-significant negative relationship with performance satisfaction. Therefore, the more an athlete perceived themselves to be resilient, the more effective they perceived their coping strategies to be over an arduous physically challenging 115 km mountain race [101].

The influence of lower levels of resilience on the psychological and behavioral outcomes appeared to be somewhat dependent on the precise nature of the profiles. For example, profiles with lower resilience (particularly when involving higher levels of anticipation and thus, awareness of threats), presented more negative outcomes such as higher anxiety, leading to a reduction in calculated risk taking. When comprised of lower anticipation (and thus a possible lack of appreciation and mitigation for risks), lower levels of resilience predicted less impulse control and less preventative behaviors in the pandemic. High impulsiveness in these individuals combined with their lower risk perception, could lead to more harmful and self-destructive behaviors [102]. Impulsive behaviors can be a response to negative emotions and low resilience to adversity [67,103] and may therefore explain why these individuals score so highly in these outcomes.

## Applied implications

Building both general and specific coping techniques that encompass or target specific processes could also provide a more individually tailored intervention. Within dynamic environments, coping can be a complex process [104], that requires a range of strategies to help deal with each domain of stress [101,105]. The present model and measurement can allow this complex process to be better understood when developing future resilience interventions. For example, each domain (physical, social, cognitive, emotional, and general) or process (anticipate, minimize, manage, and mend) could be examined individually based on the needs of the person. Improving resilience involves the use of various personal resources as well as a challenging but supportive environment [11,106]. Many of these personal qualities, skills, and coping techniques are teachable, and can potentially enhance the relevant resilience mechanisms for an adverse environment. For example, De Terte et al.'s [106] framework for building resilience is developed from Cognitive Behavioural Therapy (CBT) and describes internal and external influences on resilience. Internal factors include cognitions such as cognitive distortions, emotions, behaviours (helpful or unhelpful), and physical sensations. External factors include the environment, underpinned by support from a community and significant others [106]. One method to improve the personal qualities and internal factors influencing resilience is teaching and equipping participants with effective coping strategies [105]. These coping strategies can generally be conceived as problem-focused or emotion-focused [107]. Problem-focused coping is generally aimed at resolving a stressor directly if it's controllable. Emotion-focused coping is dealing with the effect it has on the person themselves (especially if the stressor is not controllable). These techniques could clearly map to a resilience process, allowing one to approach and anticipate adversity with a toolbox or contingency plan (minimize) of coping mechanisms to manage and mend from. As Farchi and Peled-Avram [13] discuss, this would require

a combination of having/planning potential coping resources in place (proactive), alongside tailoring and actually using and accessing them as the adversity occurs (reactive) to promote a resilience.

Furthermore, in the wider applied intervention literature on mental toughness, Bell at al. [108] set out to increase the proactive anticipatory and minimizing strategies of young cricketers while at the same time increasing their coping strategies to deal with failure as it occurred (reactive: manage). In this way, young athletes were more able to detect threats earlier (and hence minimize them earlier). From an applied resilience perspective, there are benefits to using this approach. However, if interventions are set out to increase anticipatory/minimizing strategies, then they should ensure the person has an adequate set of coping strategies (managing and potentially mending) to match them. In addition, the nature of the environment should also be considered when designing an intervention as optimal amounts of stress can lead to improvements in resilience. The process of growing from smaller, controllable adversities is known as stress inoculation [109,110]. With regards to interventions, an effective way to simulate adversity is through reward and punishment stimuli, that can relate to resilience domains. For example, limiting certain supports like social media and phone usage (social & emotional) [104], or creating demanding physical tasks (physical) [111] may be of benefit.

Another applied implication relates to resilience profiles. Assessing an individual's resilience profile will likely help practitioners better understand and predict how individuals might react to different stressful situations, thereby enabling more effective and targeted interventions to be implemented (e.g., teaching greater contingency planning and behaviors for low minimize or reducing anxiety from over-anticipation).

## Limitations, future directions, and conclusions

The findings of this study provide initial support for the reliability of the psychometric properties of the RPS. The measure establishes a new process of anticipation, as well as the cognitive domain of resilience. However, we recognise that instrument development is an ongoing process 39], and further studies are required to corroborate and validate these findings. For example, the cognitive domain would benefit from more investigation in particular; thus examining the relationship between cognitive resilience and cognitive-based tasks under stress, would provide a greater understanding of the relevance of this domain, and how cognitive resilience might protect against cognitive overload, distraction to threat, and other working memory functions [32,112]. An additional way of validating the RPS could be to examine physiological responses to stress and pain. For example, resilience can generally be linked to lower cortisol responses [27], less negative cardiovascular responses to stress [113], and being more resilient to pain [114]. Therefore, research could go some way to surmount the problems that exist with an over-reliance on self-report measures [115], social desirability bias [116], and provide further insights by linking physiological responses to certain resilience processes and domains.

By using profiling techniques demonstrated within the current study, researchers can gain more insight into resilience and its influences upon outcome variables that go beyond examining correlational relationships based within unidimensional assessments. In addition, we were able to provide some assessment of causality in terms of the influence of profiles on outcomes to supplement the cross-sectional analyses.

In terms of limitations, it is worth noting that Study 3 was conducted in a COVID-specific context, and thus the outcomes may relate to these conditions and limit the generalizability. Additionally, throughout the data collection of this study (approximately four months from February 2021), lockdowns were easing as vaccinations became more widely available. These changes may have introduced fears about the vaccine itself, reuptake of COVID-19 cases, or alleviated some psychological ill effects and stressors associated with lockdown [117]. Furthermore, we did not assess adversities that may have influenced resilience across the four-month period [99]. Overcoming or succumbing to these experiences would likely have affected perceptions of resilience across this time.

Future research could investigate antecedents of the resilience profiles themselves such as personality traits, early life experiences, stressful events and the influence of available coping resources [13] rather than focusing on outcomes. For example, given the state-like nature of resilience, it would be useful to further explore why some people transition to

different resilience profiles and others do not. Qualitative approaches might be particularly informative to uncover why some resilience profiles either remain stable or change across time (i.e., overcame or succumbed to adversity). Understanding the fluidity of resilience would further allow researchers and practitioners to plan for individual differences at an applied level and across different contexts.

Investigating RPS profiles during the COVID-19 pandemic provided a unique environment to examine profile outcomes during adversity, but future research could investigate the influence of profiles on different outcomes and in different stressful contexts. Some distinct population groups (e.g., by age, gender, or occupation across time) experience and deal with certain types of adversity differently and to varying degrees [118]. Furthermore, it should be acknowledged that the sample used in Studies 1 and 2 were limited to relatively young adults going on expeditions and students – which may limit generalizability and use of the data as a baseline of resilience. For example, those choosing to go on an expedition may already be of a higher resilience than their peers (or perhaps lack in the anticipate process) to desire such a challenge. Due to social media, some young people may have to tolerate social and emotional stress more so than an adult population. However, adults based in different workplace settings (e.g., military vs. business) would also experience a unique set of adversity relating to cognitive and physical resilience. Further investigation with resilience profiling on these population groups could address these limitations and increase our understanding of these processes in different situations.

To conclude, the current set of studies advances the literature by providing a comprehensive approach to understanding and measuring resilience as a four-stage process with proactive and reactive components across four distinct domains of functioning. We also present how this approach can be applied through profiling and present some common profiles that can occur in a population. Understanding how these profiles can potentially predict likely outcomes could help further our understanding of resilience and ensuing interventions.

## Supporting information

**S1 Appendix. Initial resilience process scale 20 items.** 20-item scale proposed during item development and examined in Study 1.
(PDF)

**S2 Appendix. Resilience process scales.** 13-item Resilience Process Scales, with vignettes separating each domain, and the scoring system for the processes.
(PDF)

**S3 Appendix. Study 1: Participant demographics, interaction effects and differences across resilience processes and domains.** Additional details and findings in Study 1.
(PDF)

**S4 Appendix. Study 2: Participant demographics, interaction effects and differences across resilience processes and domains.** Additional details and findings in Study 2.
(PDF)

**S5 Appendix. Study 3 pilot study: Hypothesized profiles.** Examination of hypothesized resilience profiles based on the literature.
(PDF)

**S6 Appendix. Study 3: Participant demographics, additional measure validation and screening.** Additional participant details, alongside validating and screening preventative behavior, negative affect, and past COVID infection, in addition to screening for extraneous influences.
(PDF)

**S7 Appendix. Study 3: Casual outcomes of resilience profiles.** Findings for causal outcomes of resilience profiles for depression, well-being, risk-taking, and impulsiveness.
(PDF)

**S8 Appendix. Supplementary references.** Additional references cited throughout supporting information materials.
(PDF)

**S1 Table. Study 1 correlations.** Table showing Study 1 correlations across resilience processes and domains.
(PDF)

**S2 Table. Study 2 correlations.** Table showing Study 2 correlations across resilience processes and domains.
(PDF)

**S3 Table. Study 3 pilot study: Fit indices.** Fit indices comparing profile solutions.
(PDF)

**S1 Fig. Study 3 pilot study: Standardized profiles.** Standardized profiles of resilience for pilot study.
(PDF)

## Acknowledgments

We would like to give special thanks to the KESS 2 Knowledge Economy Skills Scholarships and Outlook Expeditions for their support in these studies.

## Author contributions

**Conceptualization:** Joseph Anthony Pettit, Stuart Beattie, Ross Roberts, Nichola Callow.

**Data curation:** Joseph Anthony Pettit.

**Formal analysis:** Joseph Anthony Pettit.

**Funding acquisition:** Stuart Beattie, Ross Roberts, Nichola Callow.

**Investigation:** Joseph Anthony Pettit.

**Methodology:** Joseph Anthony Pettit, Stuart Beattie, Ross Roberts, Nichola Callow.

**Project administration:** Joseph Anthony Pettit, Stuart Beattie, Ross Roberts, Nichola Callow.

**Resources:** Joseph Anthony Pettit, Stuart Beattie, Ross Roberts, Nichola Callow.

**Software:** Joseph Anthony Pettit, Ross Roberts.

**Supervision:** Stuart Beattie, Ross Roberts, Nichola Callow.

**Validation:** Joseph Anthony Pettit, Stuart Beattie, Ross Roberts, Nichola Callow.

**Visualization:** Joseph Anthony Pettit.

**Writing – original draft:** Joseph Anthony Pettit.

**Writing – review & editing:** Joseph Anthony Pettit, Stuart Beattie, Ross Roberts, Nichola Callow.

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
