## [Decision Letter · Decision Letter 0]

24 Sep 2025

Dear Dr. Pettit,

Thank you for submitting your manuscript to PLOS ONE. After careful consideration, we feel that it has merit but does not fully meet PLOS ONE’s publication criteria as it currently stands. Therefore, we invite you to submit a revised version of the manuscript that addresses the points raised during the review process.

We look forward to receiving your revised manuscript.

Kind regards,

Umberto Baresi, Ph.D.

Academic Editor

PLOS ONE

Journal Requirements:

2. Please expand the acronym “KESS” (as indicated in your financial disclosure) so that it states the name of your funders in full.

“JP

Funded by KESS II, Outlook Expeditions

Grant number: BUK2133

Websites.

KESS II: https://kess2.ac.uk/

Outlook Expeditions: https://outlookexpeditions.com/

Outlook Expeditions supported in data collection for Study 1 and by extension, Study 3's pilot.”

4.   We note that your Data Availability Statement is currently as follows: All relevant data are within the manuscript and in Supporting Information files.

Additional Editor Comments:

Dear Authors,

Thank you for submitting your manuscript to our journal. After reviewing the feedback provided by the three reviewers, I believe your work deserves consideration for publication.

As noted by the reviewers, there is room for improvement, particularly in the clarity of the methods section. I kindly ask you to revise the manuscript accordingly.

Please prepare a detailed response to the reviewers’ comments, including:

- A table outlining each comment and your corresponding response.

- A revised version of the manuscript with tracked changes highlighting the modifications made.

We look forward to receiving your revised submission.

Warm regards

Reviewers' comments:

Reviewer's Responses to Questions

**Comments to the Author**

1. Is the manuscript technically sound, and do the data support the conclusions?

Reviewer #1: Yes

Reviewer #2: Yes

Reviewer #3: Yes

2. Has the statistical analysis been performed appropriately and rigorously?

Reviewer #1: No

Reviewer #2: Yes

Reviewer #3: Yes

3. Have the authors made all data underlying the findings in their manuscript fully available?

Reviewer #1: Yes

Reviewer #2: No

Reviewer #3: Yes

4. Is the manuscript presented in an intelligible fashion and written in standard English?

Reviewer #1: Yes

Reviewer #2: Yes

Reviewer #3: Yes

Reviewer #1: Major Comments:

Conceptual Clarity and Novelty:

Strengthen the reasoning for treating "anticipate" and "minimize" as separate proactive processes. Clearly contrast this with Chen et al.'s (2010) combined "anticipate" factor and Alliger et al.'s (2009) framework. Present empirical and theoretical evidence (like appraisal theory) early in the Introduction.

Clarify the "cognitive resilience" area. While it's claimed to be novel, explain how it differs from "emotional" resilience. Provide stronger theoretical support (beyond job-related examples) and describe cognitive processes (like executive function under stress) apart from emotional regulation.

Methodology and Measure Development (Studies 1 and 2):

Detail how items were reduced: The manuscript notes items were cut from 94 to 65 based on wording and similarity, then to 20 through expert ratings, and finally to 13 using BSEM. Provide:

The criteria used for expert ratings (such as inter-rater agreement statistics).

A clear justification (beyond factor loadings) for keeping items with correlated residuals that exceed the usual thresholds.

A table listing all initial items along with reasons for exclusion in the supplementary materials.

Explain the BSEM method: Briefly clarify why BSEM was more suitable than frequentist SEM (for example, for managing small cross-loadings) for PLOS ONE's wider readership. Clearly outline key prior specifications.

Include the full vignettes for each domain in the main text or a well-referenced supplement (S1 Appendix). The wording is important for understanding domain scores.

Acknowledge that the sample limitations in Studies 1 and 2 (expedition participants, UK students/community) affect generalizability. Discuss possible biases, such as the expedition group potentially having higher baseline resilience.

Profile Interpretation and Stability (Study 3):

Reconcile pilot and main profiles: The main study identified a "Low Resilience – Low Anticipate" profile (36.5%) that was not predicted in the pilot phase. Discuss possible reasons, such as the pandemic context affecting anticipation abilities or differences in samples.

Consider Profile 4 (High Resilience) size: With just 5.5% of the sample, the statistical power for comparisons involving this profile is low (evident in anxiety SE=1.03). Be cautious in interpreting its "superior" outcomes. Consider merging it with Profile 3 if justifiable or clearly state the limitations in power.

The instability in LTA transitions (for example, High Resilience to Lower Resilience) needs a more thorough discussion. Connect this to the proposed state-like aspects of resilience and potential pandemic impacts, like prolonged stress draining resources. Did specific events, such as a COVID infection, relate to these transitions?

COVID-19 Context and Outcomes:

Contextualize when data was collected: The data was gathered between Feb and July 2021, during the easing of UK lockdowns and the rollout of vaccines. Note how this specific time, characterized by shifting restrictions and new hope, may have affected perceptions of resilience and outcomes compared to the initial waves of the pandemic.

Clarify what "preventative behaviors" mean: The factor analysis is in S5 Appendix. Briefly report significant loadings or the final items in the main text for transparency. How were "Not Applicable" responses handled?

To control for pandemic impacts, consider adding a brief measure of objective or subjective pandemic hardship (like job loss or the severity of illness) as a covariate in the profile-outcome analyses to isolate resilience effects.

Discussion and Implications:

Trait vs. State: The LTA shows both stability and change. Reframe the discussion to better incorporate this, avoiding too much reliance on either trait or state views. Emphasize how profiles may reflect current resource use.

Applied Recommendations: Make the recommendations for practical use clearer. How can practitioners address specific processes (like developing "minimizing" strategies through contingency planning)? Link these to existing interventions if possible.

Cognitive Domain Validation: Emphasize that this is just a beginning proposal. Explicitly state that further research is needed to validate it against cognitive performance under stress (for example, working memory tasks).

Minor Comments:

Abstract:

State the final sample size for the main analysis in Study 3 (N=400).

Specify the key pandemic outcomes where profiles showed significant differences (for example, "Profile 1 reported higher anxiety and depression").

Tables and Figures:

For Tables 1 and 2, formatting appears corrupted in the provided text. Ensure tables are clear and legends fully explain abbreviations (like PPP, BSEM, CI) in the final submission. Report factor loadings with standard errors and credibility intervals clearly.

For Figures 1-4, make sure profiles are clearly labeled (including N per profile) and processes (Anticipate, Minimize, etc.) are distinct. Include details in the main text if possible.

Measurement:

Report internal consistency (Cronbach’s α/ω) for all RPS subscales (Anticipate, Minimize, Manage, Mend) per domain in Studies 1 and 2, not just a range.

Mention the response scale for the RPS (1-7 Likert) in the Method section.

Clarify if the PHQ-4 was scored as PHQ-2 (depression) and GAD-2 (anxiety) subscales, as implied.

Reviewer #2: Review Comments to the Author

The manuscript presents the development and validation of the Resilience Process Scales (RPS) and their application to resilience profiling during the COVID-19 pandemic. Overall, this is a well-conceived and technically sound piece of research that makes a meaningful contribution to resilience science. The multi-study design, spanning measurement development, psychometric validation, and applied latent profile analysis, is a clear strength. The work aligns with PLOS ONE’s criteria for original research, and the data broadly support the conclusions.

That said, there are a few areas where the manuscript would benefit from very minor or small revisions to enhance clarity, reproducibility, and alignment with journal requirements:

1. Methods clarity and reproducibility

While the psychometric and profile analyses are appropriate, the reporting of sampling procedures, attrition, and demographic representation could be strengthened to allow clearer evaluation of generalizability. More detail on how participants were recruited across studies, and how missing data were handled, would improve transparency.

2. Statistical reporting

The exploratory and confirmatory factor analyses are reported appropriately, but clearer justification for the specific factor retention criteria and profile model selection would be helpful. For example, why was the four-profile solution favored over alternatives? Please expand on fit indices and theoretical rationale. In the latent transition analyses, additional detail on model fit, classification accuracy, and transition probabilities would improve rigor.

3. Interpretation of findings

The conclusions are generally supported by the data, but some claims (for example, implications for clinical interventions or broad generalizations about resilience processes) should be framed more cautiously. Given the reliance on self-report and the COVID-specific context, the authors should take care not to overstate the generalizability of the findings.

4. Data availability

PLOS ONE requires full availability of the underlying data. At present, it is unclear whether item-level or raw data sufficient to reproduce analyses are accessible. Please ensure that all relevant data and code are made available via a public repository, or provide a clear justification if restrictions apply.

5. Presentation and language

The manuscript is written in clear, standard English and is intelligible. However, some sections (particularly in the Results and Discussion) could be streamlined to improve readability. A careful edit for conciseness would strengthen the presentation.

Summary recommendation:

This manuscript is promising and potentially publishable in PLOS ONE, but requires minor revision to address methodological reporting, statistical justification, data availability, and cautious framing of conclusions. The revisions suggested above would substantially improve the transparency, rigor, and clarity of the work.

Reviewer #3: I received this manuscript for review and evaluation after it had already undergone an initial round of peer review and resubmission. Therefore, my current assessment focuses specifically on aspects that I believe remain insufficiently addressed in the revised version.

The manuscript represents an important theoretical advancement in resilience research. By decomposing resilience into four processes anticipate, minimize, manage, and mend the authors provide a more nuanced and process-oriented framework compared to static trait models. The development of the Resilience Process Scales (RPS) and the empirical testing across different populations strengthen the contribution and highlight clear associations between resilience profiles and psychological as well as behavioral outcomes. The paper advances the theoretical clarity of resilience mechanisms in a commendable way.

At the same time, there are several aspects that would benefit from further elaboration:

1. Differentiation between potential and available coping resources.

The model implicitly relies on coping resources, particularly in the anticipatory and minimizing stages. However, the manuscript does not fully distinguish between potential resources (those that are theoretically present or possible) and available resources (those accessible and ready for use in real time). Clarifying this distinction, which has been emphasized in recent resilience frameworks (e.g., Farchi & Peled-Avram, 2025, The ART of Resilience), could make the model more precise and practically relevant.

2. Guidance for developing the four resilience processes.

While the manuscript convincingly identifies four key domains of resilience, it remains less clear how these processes might be cultivated or strengthened. Including preliminary considerations on pathways for development would broaden the paper’s value for interventions, training, and prevention programs.

3. Consideration of the time dimension.

The four processes (anticipate → minimize → manage → mend) are presented as a logical sequence, yet the manuscript does not explicitly situate them within different temporal phases of adversity, such as the acute stage, prolonged coping, and post-event recovery. Addressing the time dimension could enrich the model and clarify how these processes function across varying contexts.

4. Application to intervention, prevention, and rehabilitation.

Although the discussion notes that resilience profiles may guide future interventions, the practical implications remain underdeveloped. Extending the discussion to illustrate how the model might inform strategies for acute intervention, preventive measures, or long-term rehabilitation would significantly strengthen the manuscript’s applied relevance.

In conclusion, this is a strong and valuable theoretical contribution that provides a clearer framework for mapping resilience. To maximize its impact, I encourage the authors to expand the discussion on resources, development of processes, the time dimension, and applied implications. Doing so would help bridge the gap between theoretical insight and practical application, enhancing both scholarly and real-world relevance.

**Do you want your identity to be public for this peer review?** For information about this choice, including consent withdrawal, please see our Privacy Policy

Reviewer #1: **Yes:** Alejandro Botero Carvajal

Reviewer #2: **Yes:** Seth Henry Britton Saeugling

Reviewer #3: **Yes:** Moshe U. Farchi

---

## [Author Response · Author response to Decision Letter 1]

8 Dec 2025

We would like to thank all reviewers for their detailed and helpful comments. We have made a number of changes to address the feedback given and we hope you find our manuscript much stronger as a result.

In the attached file "Response to Reviewers", we have created tables for each reviewer's comments and our responses/changes to each and where they can be found on the revised manuscript (with track changes) for ease. We appreciate the time and effort you have all put in for this, and hope we have made this process as smooth as possible.

---

## [Editor Report · Decision Letter 1]

8 Jan 2026

Mapping resilience: Development of the Resilience Process Scales (RPS) and resilience profiles during adversity

PONE-D-25-16922R1

Dear Dr. Pettit,

We’re pleased to inform you that your manuscript has been judged scientifically suitable for publication and will be formally accepted for publication once it meets all outstanding technical requirements.

Kind regards,

Umberto Baresi, Ph.D.

Academic Editor

PLOS One

Additional Editor Comments (optional):

Dear Authors,

I would like to thank you for the time and effort you devoted to addressing the comments.

I believe that this manuscript has improved substantially and is now ready for publication in PLoS One.

Thank you.
---

## [Editor Report · Acceptance letter]

PONE-D-25-16922R1

PLOS One

Dear Dr. Pettit,

I'm pleased to inform you that your manuscript has been deemed suitable for publication in PLOS One. Congratulations! Your manuscript is now being handed over to our production team.

Kind regards,

on behalf of

Dr. Umberto Baresi

Academic Editor

PLOS One